1    tTitel page:

# Ocean Alkalinity Enhancement impacts: Regrowth of marine microalgae in alkaline mineral concentrations simulating the initial concentrations after ship-based dispersions

Stephanie Delacroix[1], Tor Jensen Nystuen[1], August E Dessen Tobiesen[1], Andrew L. King[1],
Erik Höglund[1],
[1]Norwegian Institute for Water Research, Økernveien 94, 0579 Oslo, Norway
*Corresponding author: Stephanie Delacroix (stephanie.delacroix@niva.no)*

Manuscript:

# Ocean Alkalinity Enhancement impacts: Regrowth of marine microalgae in alkaline mineral concentrations simulating the initial concentrations after ship-based dispersions

Stephanie Delacroix[1], Tor Jensen Nystuen[1], August E Dessen Tobiesen[1], Andrew L. King[1], Erik Höglund[1],

[1]Norwegian Institute for Water Research, Økernveien 94, 0579 Oslo, Norway

*Corresponding author: Stephanie Delacroix (stephanie.delacroix@niva.no)*

**Abstract**

Increasing the marine $CO_2$ absorption capacity by adding alkaline minerals into the world's oceans is a promising marine carbon dioxide removal (mCDR) approach to increase the ocean's $CO_2$ storage potential and mitigate ocean acidification. Still, the biological impacts of dispersion of alkaline minerals needs to be evaluated prior to its field deployment, especially the impacts of the initial discharge causing local and temporary extreme alkalinity/pH changes. In this study, the toxicity effect on marine microalgae of two commonly used alkaline minerals, calcium hydroxide ($Ca(OH)_2$) and magnesium hydroxide ($Mg(OH)_2$), by adding the same equivalent molar concentration of hydroxyl ions. Cultures of marine green microalgae *Tetraselmis suecica* were exposed to $Ca(OH)_2$ or $Mg(OH)_2$, in concentrations mimicking the initial high concentrations following a dispersion scenario from a ship. A short-term exposure with high alkaline mineral concentration called "dispersion phase" was followed by a dilution step and a "regrowth" phase over six days. There was no detectable effect of $Mg(OH)_2$ treatment on algae growth either after the dispersion phase or during the regrowth phase, compared to control treatments. The $Ca(OH)_2$ treatment resulted in very few living algal cells after the dispersion phase, but a similar growth rate was observed during the regrowth phase as was for the $Mg(OH)_2$ and control treatments. Standardized whole effluent toxicity (WET) tests were carried out with a range of $Mg(OH)_2$ concentrations using a sensitive marine diatom, *Skeletonema costatum,* which confirmed the relative low toxicity effect of $Mg(OH)_2$. Similar biological effects were observed on natural microalgae assemblages from a local seawater source when applying the same $Mg(OH)_2$ concentration range and exposure time used in the WET tests. The results suggest that $Mg(OH)_2$ is relatively safe compared to $Ca(OH)_2$ with respect to marine microalgae.

## 1 Introduction

It is widely recognized that reducing carbon dioxide emissions is not sufficient to accomplish the goals of the Paris agreement of 2015, limiting global warming and ocean acidification (Pathak et al., 2022). Accordingly, there is an urgent need for additional carbon dioxide removing approaches. Many different marine dioxide carbon removal (mCDR) approaches are currently under evaluation (GESAMP, 2019), including artificial upwelling/downwelling, nutrient fertilization, deep sea storage, electrochemical ocean carbon dioxide removal, macroalgal/microalgal cultivation, marine ecosystem restoration, and ocean alkalinity enhancement (OAE). In general, the principle of some of these approaches is based on acceleration of the natural process of absorption and long-term storage of the excess atmospheric carbon dioxide by the ocean (Siegel et al., 2021, NASEM, 2021). Among them, OAE has been put forward as one of the most promising approaches, because the acidification remediation process itself triggers the reduction of the atmospheric carbon dioxide level (Renforth and Henderson, 2017). Hence, when the aquaeous carbon dioxide deficit, generated by the addition of alkaline mineral, returns to the initial equilibrium with atmospheric carbon dioxide, the final pH still remains slightly higher than the initial pH, while calcite (most stable polymorph of calcium carbonate CaCO3) level and aragonite (crystal structure of calcium carbonate) saturation state are elevated. The aragonite saturation state is commonly used to track ocean acidification (Qing-Jiang et al., 2015). The most studied alkaline minerals for OAE approaches are limestone ($CaCO_3$), olivine $(Mg,Fe)_2SiO_4$, sodium hydroxide (NaOH) and calcium hydroxide ($Ca(OH)_2$) (DOSI, 2022). While the latter mineral has been evaluated for large scale application on the Mediterranean Sea (Butenschön et al., 2021), a large-scale study involving field deployment of olivine in coastal waters off New York, USA is currently being performed (Tollefson, 2023). Magnesium hydroxide has also recently been studied (Yang et al., 2023; Hartmann et al., 2022). Its relatively low water solubility allows it to be added in a larger amount without reaching harmful pH levels (Tollefson, 2023) and will potentially increase the durability of the alkalinization effect. Following this, in addition to raw material source scalability (Caserini et al., 2022), alkalinization efficiency and solubility are important criteria of OAEs (Hartmann et al., 2022; Ilyina et al., 2013). Moreover, the effects on the aquatic environment need to be considered, including the biological impact of the initial discharge of high alkaline mineral concentrations upon dispersion causing local and temporary extreme alkalinity/pH changes. Accordingly, Bach et al., (2019) and Burns and Corbett (2020) pointed out that before approval of the alkaline mineral dispersion at global scale, a risk assessment of the toxicity effect of the alkaline minerals on marine organisms must be performed. Thus, it is crucial to consider not only the toxicity effect, if any, of the final low alkaline mineral concentration after expected final dilution into ocean, but also the potential initial toxicity effect of the initial hot spot discharge of the alkaline mineral on local organisms. These discharges upon dispersion might be local and temporary, but it is important to consider that they would be applied at a global scale. These local and temporary effects will potentially include increased cation levels ($Mg^{2+}$ and $Ca^{2+}$), increased bicarbonate and carbonate ions, pH increase or decrease of dissolved carbon dioxide. Perturbations that potentially form impact hotspots, affecting phytoplankton species composition and growth, resulting in impacts higher up in the food chain (Bach et al., 2019). Biological impacts will strongly depend on the spatial and temporal scale of alkaline mineral dispersion, and studies must therefore use realistic alkaline mineral dispersion scenarios.

In this study, the biological impact of initial and temporary discharge of $Mg(OH)_2$ concentrations expected from dispersion from a moving ship was compared to $Ca(OH)_2$ on marine microalga. This was done by exposing cultured *Tetraselmis suecica* to the above alkaline minerals. The toxicity of $Mg(OH)_2$ was then further investigated

by using a sensitive microalgal species, in a recognized and standardized whole effluent toxicity (WET) test with
cultured diatom *Skeletonoma costatum*. Additional experiments were performed for further toxicity assessment
of $Mg(OH)_2$ on a natural microalgal assemblage from local seawater.

## 2 Methods

The study was performed in three steps. In the first step, the toxicity effect was studied by exposing marine alga
to alkaline minerals in successive concentrations mimicking dispersion from a moving ship. These experiments
were carried out with *Tetraselmis suecica*, a standard test organism in toxicity studies (Ebenezer et al., 2017; Li
et al., 2017; Seoane et al., 2014; Vagi et al., 2005). In the second step, toxicity effects of the alkaline minerals
were verified by a standardized WET ecotoxicology assay with *Skeletonoma costatum*, a more sensitive marine
algal species (Petersen et al., 2014, Wee et al., 2016), by using the recognized 72 hours growth inhibition test
(ISO 10253:2016). In the third step, the toxicity effect was studied by exposing a natural assemblage of marine
algal species from the Oslofjord, Drøbak, Norway to similar $Mg(OH)_2$ concentrations used in the WET tests. All
experiments were carried out in non-airtight containers to allow ambient $CO_2$ to re-equilibrate with seawater used
for the experiments.

### 2.1 Exposure of *Tetraselmis suecica* to simulated dispersion of alkaline minerals from a moving ship

The expected distribution of a slurry of $Mg(OH)_2$ during its dispersion from the ship's discharge point on the
surface of the oceans was determined utilizing computational fluid dynamic (CFD) models (FORCE Technology
Inc., Denmark) and the Bottom RedOx Model (BROM) (Yakushev et al., 2017). In those models, both the forced
and natural mixing effects of the $Mg(OH)_2$ by the ship's propeller and physical oceanic processes (as waves,
convection, currents, etc.), respectively, in the ship's wake were simulated with different scenarios, including
propeller motion, velocity of tangential ocean currents, $Mg(OH)_2$ slurry discharge rate/dissolution rate/settling
rate, ship size and ship speed. Dilution was observed with an immediate minimum dilution rate of 1/1000 within
2 minutes after injection, followed by an additional minimum dilution rate of 1/7000 during the next 5 hours and
a final minimum dilution rate of 1/154000 during the following next 5 hours. Moreover, the tonnage capacity and
operating costs of a ship were also considered together with a final $Mg(OH)_2$ concentration target of < 1 mg $L^{-1}$.
Taken together, this suggested that the dispersion rate of 500 kg $s^{-1}$ would be the most realistic applicable scenario.
From this dispersion rate, it was concluded that marine organisms would be exposed to < 100 g $L^{-1}$ approximately
for less than one hour followed by a dilution to <10 mg $L^{-1}$ over a period of 10 hours.
To investigate biological impact of $Mg(OH)_2$ and compare it with $Ca(OH)_2$, cultures of *Tetraselmis suecica* were
exposed to these three alkaline minerals during a simulated dispersion phase (as described above) followed by a
regrowth phase (Fig 1). In the dispersion phase, 30 mL of *Tetraselmis suecica* cultures (see further down), in
exponential growth with a cell density range within $2.6 \times 10^5$ - $1.4 \times 10^6$ cells $mL^{-1}$, were exposed to the alkaline
minerals in 50 mL glass beakers with continuous mixing at approximately 300 rpm with a magnetic stirrer (VELP
Scientifica) for 1 hour. To achieve similar concentrations of hydroxide ions in the different alkaline mineral
treatments, algae were exposed to either 100 g $L^{-1}$ (or 1.7 M) of $Mg(OH)_2$ or 127 g $L^{-1}$ (or 1.7 M) of $Ca(OH)_2$
(Fig.1).

In the regrowth phase, a subsample from each exposure media was diluted by 10,000 in local seawater and algal cell density was monitored for 6 days. The dilution was performed by mixing 0.25 mL subsample to 2.5 L ambient 60 m deep seawater from the Oslofjord (Fig.1). The diluted subsamples were incubated in 3 L glass beakers in a 20°C temperature-controlled climate room with 24h light (2x 21W Philips Pentura Mini) and continuous mixing with a magnetic stirrer (VELP Scientifica; 100 rpm approximately). The measured light intensity was within 20-60 $\mu$mol photons $m^{-2}$ $s^{-1}$. As the beakers were left uncovered, evaporated water volume was replaced every 24h (except for week-end period) by an equivalent volume of ultrapure water. Effects of each alkaline mineral were investigated in triplicates, including both the exposure and regrowth phases; resulting in total of nine bioassays which were conducted in NIVA's laboratory in Oslo between November 2021 and January 2022. Each bioassay study was conducted with one or two alkaline minerals in parallel and were repeated three times for each alkaline mineral with new cultures of *Tetraselmis suecica*, except for two of the NaOH studies which were started on the same day from the same algal culture. In addition, control bioassays excluding the addition of alkaline minerals were performed in parallel to each alkaline mineral exposure including a dispersal phase followed by a regrowth phase.

The ambient Oslofjord seawater was unfiltered and unsterilized water collected from 60 m depth just outside of NIVA's marine research station located at Drøbak, 40 km south of Oslo. The water quality of this seawater is stable year-round with a temperature of approximately 7°C. This water is representative of ocean regions; i.e. rich in oxygen but poor in inorganic and organic contents, with 0.7 mg C $L^{-1}$ of particulate carbon (POC), 1.1 mg C $L^{-1}$ of dissolved organic carbon (DOC), 6 mg $L^{-1}$ of total suspended solids (TSS) and very low biological load with < 1 cell $mL^{-1}$ of algae and less than 500 CFU $mL^{-1}$ of heterotrophic bacteria.

## Dispersal phase

| Mg(OH)₂ exposure (100 g L⁻¹) | Ca(OH)₂ exposure (127 g L⁻¹) | Control (without alkaline mineral) |
|---|---|---|

**Mg(OH)₂ exposure (100 g L⁻¹)**

3.00 g Mg(OH)₂ powder + 30mL Tetraselmis culture

Fast mix for 1 hour

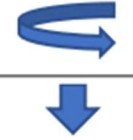

**Ca(OH)₂ exposure (127 g L⁻¹)**

3.82 g Ca(OH)₂ powder + 30mL Tetraselmis culture

Fast mix for 1 hour

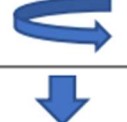

**Control (without alkaline mineral)**

30mL Tetraselmis culture

Fast mix for 1 hour

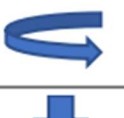

## Regrowth phase after 10,000x dilution

**10 mg L⁻¹ Mg(OH)₂**

2.5L 60m SW + 0.25mL exposed culture

24h light and 20°C

Slow mix for 6 days

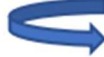

**12.7 mg L⁻¹ Ca(OH)₂**

2.5L 60m SW + 0.25mL exposed culture

24h light and 20°C

Slow mix for 6 days

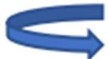

**Control (without alkaline mineral)**

2.5L 60m SW + 0.25mL control culture

24h light and 20°C

Slow mix for 6 days

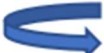

**Figure 1. Schematic illustration of the experimental set-up including the dispersion phase in 50 mL glass beakers followed by the dilution step and the regrowth phase of the exposed algal cells in 3 L glass beakers.**

Before exposure, the algae were collected from 1 L laboratory cultures of *Tetraelmis suecica* (NIVA-3/10; Norwegian Institute for Water Research, Oslo, Norway). At first, a 50 mL algal culture was prepared by semi-static cultivation in a 100 mL glass flask with 50 mL of autoclaved 20% Z8 culture medium with addition of vitamins (Kotai, 1972). The medium culture was inoculated with 5-10 mL of the *T. suecica* culture from NIVA's algal culture collection. The culture was incubated for ~1 week with fluorescent light tubes giving 20-60 µmol photons m⁻² s⁻¹, provided by cool-white fluorescence lamps (TLD 36W/950, Philips, London, UK), on an Infors Multicrom 2 incubator shaker (Infors AG, Bottningen, Switzerland) at 20 ± 2°C, with orbital shaking at 90 rpm. After incubation, the culture was used for the inoculation of the 1L culture, except for ~10 mL which was held

back to start a new 50 mL culture by adding 40 mL of freshly prepared Z8 medium in same culture conditions as described above. The 1 L culture was prepared by static cultivation with 1 L autoclaved 20 % Z8 medium with addition of 1 mL $L^{-1}$ vitamins in 2-liter glass culture bottles. Approximately 40 mL of the 50 mL stock culture was added to 1 L of medium. The culture was exposed to fluorescent light tubes of 20-60 $\mu M$ $m^{-2}$ $s^{-1}$ and placed in a 20°C temperature-controlled room for approximately one week.

The culture medium was prepared at least 24 h before usage to allow the equilibrium of media components. The 20% Z8 culture medium was made by mixing 0.2 L of Z8 medium into 0.8 L seawater, and shortly aired with $CO_2$ (< 1 min) to avoid precipitation of salts during autoclaving. The seawater was pasteurized seawater collected from 60 m depth in the Oslofjord. The medium was autoclaved for 15 minutes at 121°C. 1 mL $L^{-1}$ of vitamins stock solution was added to the 20% Z8 medium (Kotai, 1972).

The studied alkaline minerals were magnesium hydroxide (CAS number: 1309-42-8), calcium hydroxide (CAS number: 1305-62-0) and sodium hydroxide (CAS number: 1310-73-2); all with ≥97.0% purity. Magnesium hydroxide (Batch No. 18417-01A) was provided by Negative Emission Materials, Inc. via a factory in Canada producing the mineral by hydrometallurgy process and purification from natural magnesium silicate. The two other alkaline minerals were purchased from Sigma-Aldrich (United Kingdom).

The density of living *Tetraselmis suecica* was determined using the double staining method with fluorescein diacetate (FDA) and 5-chloromethylfluorescein diacetate (CMFDA) (NSF, 2010). This double staining method, FDA/CMFDA, is based on the validation work of the US Navy Research Laboratory to distinguish between living and dead cells after disinfection by a ballast water treatment (Steinberg et al., 2011). This viability method is the only one recognized by both the International Maritime Organization (IMO) and the United States Coast Guard (USCG) for approval of ballast water discharge from 70,000 commercial ships at a global scale (USCG, 2012, IMO, 2018).

The following staining protocol was used: A 2.5 mM CMFDA stock solution was prepared by dissolving 1 mg of CMFDA in 0.86 ml DMSO (Dimethyslsulphoxide). It was then divided into 50 µl batches and stored at -20 °C. The 5 mM FDA stock solution was prepared by dissolving 10 mg FDA in 4.8 ml DMSO. The FDA stock solution was divided into 100 µl batches and stored at -20 °C. For each analysis, a 4 ml subsample was collected and 4 µl of 10% HCl was added, bringing the pH back to approximately 8 prior to staining. 4 µl of each stock solution was added to each subsample, resulting in final concentrations of 2.5 µM CMFDA and 5µM FDA. The subsamples were then incubated in darkness for 10 minutes, after which they were loaded into 1 ml Sedgewick-Rafter counting chambers etched with 1-mm 2 grids. Chambers were examined at 100x magnification using compound epifluorescent microscopes with standard blue light excitation (480 nm) and green bandpass emission (530 nm) filter cubes. Samples were counted within a 45-minute period after incubation. The stained *Tetraselmis suecica* cells were counted in triplicate (3x 1 mL) The untreated algal samples without alkaline mineral were used as positive controls. Both *T. suecica* and local diatoms are nearly 100% stainable with these stains according to our 15 years of experience with this method in our local seawater. Samples treated with sodium hydroxide (NaOH) to increase the pH to approximately 14 were used as negative controls. No fluorescence could be observed in the negative controls, indicating an instant kill effect of the algal cells.

Temperature, salinity and pH in the bioassays were measured in-situ by using a calibrated handheld WTW Multimeter (WTW Multi 3620 IDS/3420 IDS displayer) with a conductivity probe (TetraCon 925 Xylem) and a pH-electrode (SenTix 945P). The three-point calibration method with Hamilton pH-buffer solutions (4, 7 and 10) was used for the calibration of the pH electrode, according to WTW instructions. The temperature in the test waters varied within a range of 18-23°C for all experiments during the 6 days of regrowth phase as all experiments were conducted at room temperature. The same temperature was registered in the alkaline test waters compared to the corresponding control waters. The salinity of the test waters, with or without alkaline mineral, was around 32-33 PSU at the start of the 6 days regrowth phase for all experiments. The salinity stayed relatively stable for most of the regrowth phase, except for the last day with an increase up to 35-36 PSU on average. This increase was due to the evaporation of the test water at room temperature during the week-end period included at the end of the 6 days of experimentation.

## 2.2 Whole Effluent Toxicity (WET) test

The WET test consisted of a marine algal growth inhibition test of 72 hours performed by NIVA's ecotoxicity laboratory according to NIVA's standard procedure which is based on International Standard ISO 10253: Water Quality – Marine algal growth inhibition test with *Skeletonema costatum* and *Phaeodactylum tricornutum*. In this study, the diatom *S. costatum* (NIVA-strain BAC 1) was used as test organism.

A 100 mg $L^{-1}$ $Mg(OH)_2$ sample was diluted by using a modified ISO 10253 media, except that no Fe-EDTA stock solution was added, as the tested compound $Mg(OH)_2$ showed to be affected by the presence of EDTA causing precipitation of $Mg(OH)_2$. A preliminary study was made to verify the microalgal growth in this modified media. Although less growth was observed when compared to normal ISO 10253 media, the specific daily growth rate was still greater than 0.9 $d^{-1}$, which was considered as valid. A total of six concentrations of $Mg(OH)_2$ was tested (1, 10, 25, 50, 75 and 100 mg $L^{-1}$). The test was performed with 15 mL samples in 30 mL glass vials. Each concentration was tested in triplicate with 6 replicates for each control (one control set with normal ISO 10253 and another control set with modified ISO 10253); same number of replicates for analysis of blank samples but without microalgae added.

All samples were inoculated with 5 x $10^6$ cells $L^{-1}$ *of S. costatum* from an exponentially growing laboratory culture and incubated on a shaking table at 20±1°C under continuous illumination of 63 $\mu M$ $m^{-2}$ $s^{-1}$ of photosynthetic active radiation (PAR).

The cell density was determined by FDA and CMFDA double staining and fluorescence at 645 nM inSpectraMax iD3 microplates after approximately 24, 48 and 72 hours (±2h). The fluorescence measurements were directly correlated to the algal density as a correlation factor ($r^2$) of 1 between the measured fluorescence and the cell density was calculated. The fluorescence values of the exposed samples without algae (blanks) were measured to investigate potential biases caused by effect of the tested substance on the fluorescence readings. As no such effects were detected, no further transformation of data was necessary.

The temperature, pH and salinity were measured in-situ at the beginning and at the end of each WET test. The temperature varied from 19.9 to 20.3°C for both WET tests. The pH at the start of the experiment varied from 8.089 to 9.376 in all vials for both tests, with increasing pH for increasing $Mg(OH)_2$ concentrations as expected.

The pH at the end of the experiment varied from 8.270 to 8.540 in all vials for both tests. The salinity was stable with 32-35 PSU in all vials during the entire experiment for both tests.

**2.3 Natural assemblage of ambient marine algal test**

For the preparation of the ambient algal culture, either a 25 L grab-sample from the surface water of Oslofjord was directly used for the test or a 2 L subsample was mixed to 2 L of 60 m deep seawater from Oslofjord for further algal growth. For growth, the culture was incubated in a 5 L glass beaker in a climate-room at 20°C and with constant light from fluorescent light tubes of 20-60 $\mu M$ $m^{-2}$ $s^{-1}$ for four days. The total density of algal cells in the culture after incubation was approximately 1000 cell $mL^{-1}$. 500 mL of the culture was then mixed, in a 2 L glass beaker with a magnetic stirrer at approximately 90 rpm, added to 1500 mL of a prepared $Mg(OH)_2$ suspension resulting in $Mg(OH)_2$ concentrations of 1, 10, 25, 50, 75 and 100 mg $L^{-1}$ and initial algal density of approximately 125-250 cell $mL^{-1}$. The $Mg(OH)_2$ suspensions were prepared by mixing 2.7 mg, 27 mg, 66 mg, 133 mg, 200 mg or 270 mg of $Mg(OH)_2$ in 1.5 L of unfiltered 60 m seawater from Oslofjord, with a magnetic stirrer (300 rpm) over the night prior test start. The final solutions were slowly mixed continuously with a magnetic stirrer at approximately 90 rpm, in a climate room at 20°C and with constant light from fluorescent light tubes of 20-60 $\mu mol$ photons $m^{-2}$ $s^{-1}$ for 72 hours. The water quality and algal density was monitored daily in each beaker, the same methods described in 2.1. Moreover, cell count and viability were quantified using the same protocol as in 2.1., with florescence measured at 645 nm. For the control treatments, 500 mL of the ambient algal culture was mixed with 1.5 L of unfiltered 60m deep seawater from Oslofjord, without $Mg(OH)_2$, and incubated as described above. Those tests were carried out on different weeks. Therefore, different control treatments applied for 1-10 mg/L $Mg(OH)_2$ treatments, 50-75 mg/L $Mg(OH)_2$ treatments and 100 mg/L $Mg(OH)_2$ treatment (see Appendix C). Aliquots from the 100 mg $L^{-1}$ treatment were collected from the initial timepoint and final timepoint (t=3 d) for microscopy-based assessment of community composition by taxa.

**2.4 Data analysis**

Effects on *T. suecica* cell survival with $Ca(OH)_2$, and $Mg(OH)_2$ in simulated dispersions from a moving ship were analyzed with a Student's t-test with type of alkaline mineral as independent grouping variable and % survival compared to control treatments after the regrowth phase as the dependent variable. Data were log transformed to obtain similar variation between groups.

In the WET test, the growth rate of *S. costatum* in each $Mg(OH)_2$ sample was calculated from the logarithmic increase of cell density from start to 72 hours, and expressed as percentage of the growth rate of control samples. The concentrations causing 50% growth inhibition ($EC_{50}$) were calculated using a non-linear regression analysis of the growth rate versus log cell concentration of control water (Hill, 1910; Vindimian et al. 1983). The non-observed effect concentration (NOEC) and the lowest observed effect concentration (LOEC) were calculated using Dunnett's test/ t-test for non-homogenous variance and Williams Multiple Sequential t-test for homogenous variance.

Effects of $Mg(OH)_2$ on the natural marine algal assemblage was investigated by dividing the different exposure concentrations (1, 10, 25, 50, 75 and 100 mg $L^{-1}$) within two groups based on the LOEC (25 mg $L^{-1}$) from the WET test. This resulted in one low concentration group (1, 10 and 25 mg $L^{-1}$) and one high concentration group (50, 75 and 100 mg $L^{-1}$). The difference in % survival compared to control treatment between the high and low

concertation groups was investigated by a Student's t-test.  This approach, with three replicates in each group,
allowed us to investigate effects of increased MgOH$_2$ concentrations.

## 3 Results

### 3.1 Exposure of *Tetraselmis suecica* to simulated dispersion of alkaline minerals from a moving ship

There were significant differences in living cells of *Tetraselmis suecica* (% survival compared to control
treatments; Table 1) between the alkaline minerals in the end of the regrowth phase (Student's t-test; t=9.4,
P<0.0001), which were reflected in both the dispersion and the regrowth phases. At the start of the regrowth phase,
the surviving cell densities in the Mg(OH)$_2$ treatments were similar to the ones observed in control treatment,
while only one living cell was observed in one of the Ca(OH)$_2$ treatments (Day 0; Table 1). In the Mg(OH)$_2$ and
Ca(OH)$_2$ treatments, algal cell densities increased during the regrowth phase (Day 1-6; Table 1). At the end of the
regrowth phase, the algal cell densities in the Mg(OH)$_2$ treatments were similar as in control treatments, while the
algal cell densities in Ca(OH)$_2$ treatments showed lower values than in control treatments (Fig.2).
**Table 1. Densities of living *Tetraselmis suecica* (cell mL$^{-1}$) and their relation to control treatment (% Contr.) during the**
**regrowth phase of a bioassay mimicking dispersion of the alkaline minerals Mg(OH)$_2$ or Ca(OH)$_2$ from a ship. Before**
**the regrowth phase, algae were exposed to either 100 g L$^{-1}$ Mg(OH)$_2$ or 127 g L$^{-1}$ Ca(OH)$_2$  (achieving similar amount**
**of hydroxide in the different alkaline mineral  suspensions) for 1h. After this, subsamples from each treatment were**
**diluted 10 000 times and algae growth were studied during a 6-day regrowth phase. Each alkaline mineral was assayed**
**in triplicates. Values at day zero corresponds to 1h after dilution and effects of each alkaline mineral was investigated**
**in triplicates.**

| | Mg(OH)$_2$ | | | | | | Ca(OH)$_2$ | | | | | |
|---|---|---|---|---|---|---|---|---|---|---|---|---|
| | Replicate 1 | | Replicate 2 | | Replicate 3 | | Replicate 1 | | Replicate 2 | | Replicate 3 | |
| Day | Cells ml$^{-1}$ | % Contr. | Cells ml$^{-1}$ | % Contr. | Cells ml$^{-1}$ | % Contr. | Cells ml$^{-1}$ | % Contr. | Cells ml$^{-1}$ | % Contr. | Cells ml$^{-1}$ | % Contr. |
| 0 | 27 | 84 | 30 | 97 | 82 | 53 | 0 | 0 | 0 | 0 | 1 | 2.9 |
| 1 | 40 | 62 | 64 | 145 | 84 | 53 | 1 | 0.66 | 0 | 0 | 1 | 2.5 |
| 2 | 72 | 63 | 129 | 168 | 256 | 64 | - | | 0 | 0 | 3 | 3.5 |
| 3 | 101 | 72 | 249 | 199 | - | - | 6 | 0.60 | 0 | 0 | 4 | 3.6 |
| 6 | 1040 | 84 | 1533 | 263 | 6217 | 128 | 56 | 0.68 | 1 | 0.11 | 29 | 2.3 |


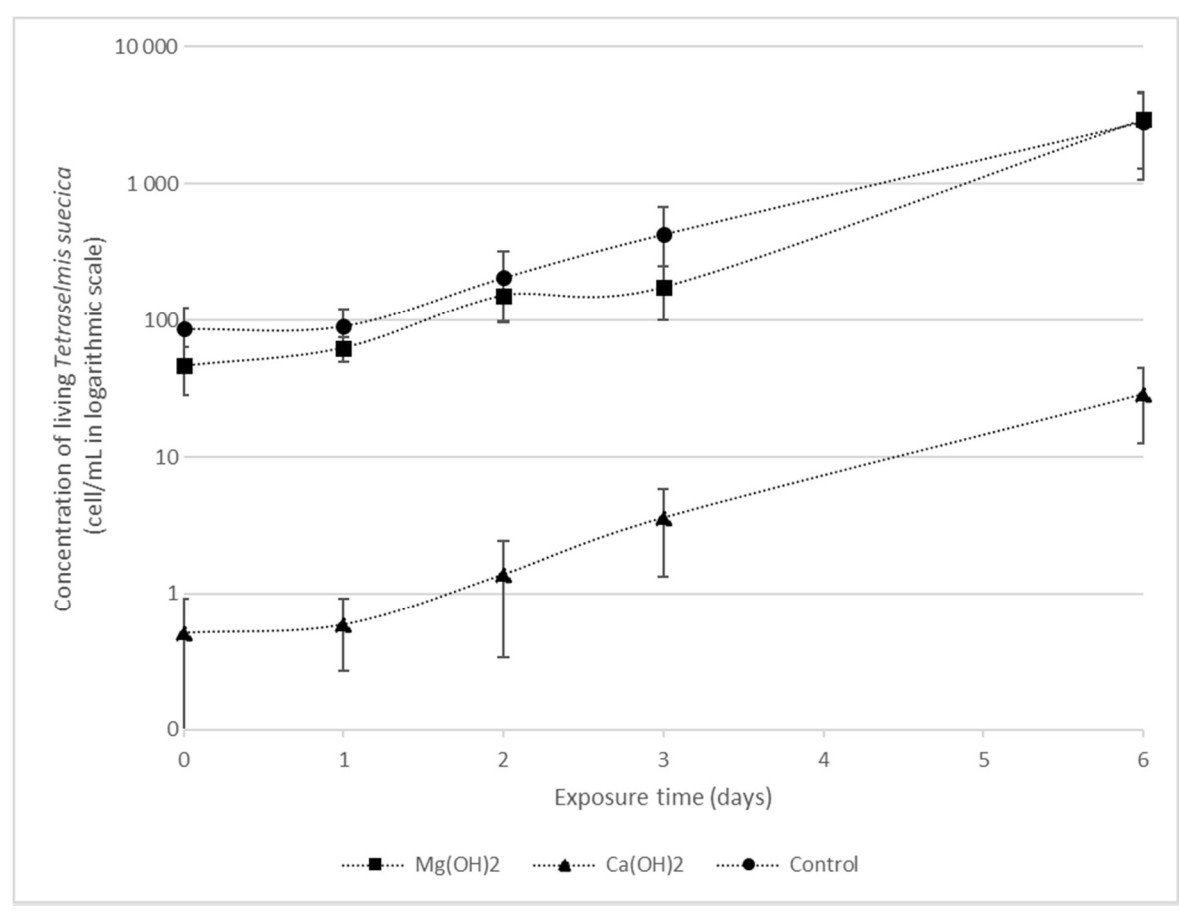


**Figure 2. Densities of living Tetraselmis suecica (cell mL⁻¹) during the regrowth phase of a bioassay mimicking dispersion of the alkaline minerals Mg(OH)₂ or Ca(OH)₂ from a ship. Before the regrowth phase, algae were exposed to either 100 g L⁻¹, Mg(OH)₂ or 127 g L⁻¹ Ca(OH)₂ (achieving similar concentrations of hydroxide ions in the different solutions) for 1h. After this, subsamples from each treatment were diluted 10 000 times and algae growth were studied during the 6 days regrowth phase.**


The pH in the control treatments were around 8.0-8.2 during the regrowth phase (Fig. 3). While alkaline mineral
treatments resulted in elevated pH (~ 8.5) at day one after dilution step. Where upon, pH decreased and reached
similar values as control treatments in day 3 for all alkaline mineral treatments (Fig. 3).

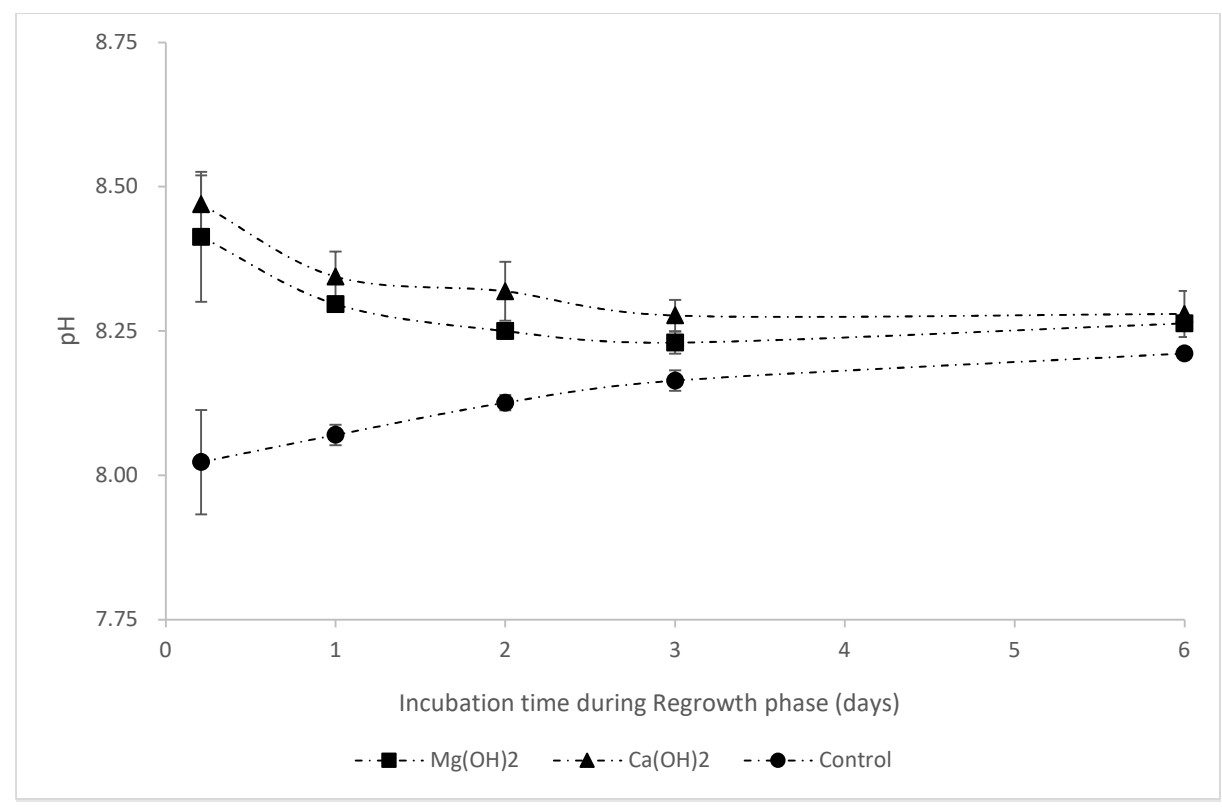


**Figure 3. pH during the regrowth phase in a bioassay mimicking dispersion of the alkaline minerals Mg(OH)$_2$ or**
**Ca(OH)$_2$ from a ship. Before the regrowth phase, algae were exposed to either 100 g L$^{-1}$ Mg(OH)$_2$ or 127 g L$^{-1}$ Ca(OH)$_2$**
**(achieving similar concentrations of hydroxide ions in the different alkaline mineral solutions) for 1h. After this,**
**subsamples from each treatment were diluted 10 000 times to achieve the following concentrations during the regrowth**
**phase; 10 mg L$^{-1}$ Mg(OH)$_2$ or 12.7 mg L$^{-1}$ Ca(OH)$_2$.**

304

### 3.3 WET tests

The results of the lowest observed effect concentration (LOEC) and the non-observed effect concentration
(NOEC) of Mg(OH)$_2$ were similar in both WET tests; with 50 mg L$^{-1}$ and 25 mg L$^{-1}$ Mg(OH)$_2$, respectively. The
Mg(OH)$_2$ concentration causing 50% algal growth inhibition was close to 100 mg L$^{-1}$ in both tests; within a range
of 82-111 mg L-1 (Table 2).

**Table 2. Results of the duplicate Whole Effluent Toxicity (WET) tests (WET tests 1 and 2) for three endpoints (EC$_{50}$,**
**LOEC and NOEC) after 72 hours exposure of the marine microalgae *Skeletenoma costatum* with a total of six different**
**concentrations of magnesium hydroxide (1, 10, 25, 50, 75 and 100 mg L$^{-1}$). Those concentrations were prepared by**
**diluting an initial Mg(OH)$_2$ solution in the algal culture medium, prior to algal inoculation. The initial solution was a**
**freshly prepared 1 L suspension of 100 g L$^{-1}$ Mg(OH)$_2$ in ambient 60m deep seawater from Oslo fjord. EC50:**
**concentration causing 50% algal growth inhibition. LOEC: lowest observed effect concentration. NOEC: non-observed**
**effect concentration (NOEC).**

| | WET tests, Mg(OH)$_2$ (mg L$^{-1}$) | |
|---|---|---|
| Endpoint | 1 | 2 |
| EC$_{50}$ | 111 | 82 |
| LOEC | 50 | 50 |
| NOEC | 25 | 25 |

317

### 3.4 Natural assemblage of ambient marine algal species

There was a significant difference in algal survival between the low concentrations group (1, 10 and 25 mg L$^{-1}$ $Mg(OH)_2$) and the high concentrations group (50, 75 and 100 mg L$^{-1}$ $Mg(OH)_2$) after three days of exposure ($t_{(4)}$=-5.8, P<0.01; Table 3). The analysis of the algal biodiversity composition in the 100 mg L$^{-1}$ $Mg(OH)_2$ suspension showed that the dominant surviving species were diatoms, including *Skeletonoma spp.,* with 80% and 94% of the total on Day 0 and Day 3, respectively. The biodiversity composition of the natural algal assemblage in beginning and at the end of the experiment for the 100 mg L$^{-1}$ $Mg(OH)_2$ treatment is given in Table 4.

**Table 3. Densities of living ambient algal cells (cell mL$^{-1}$), and their survival in percentage compared to corresponding control water without $Mg(OH)_2$ (% Contr.), during 3 days of exposure to six different concentrations of $Mg(OH)_2$ (1, 10, 25, 50, 75 and 100 mg L$^{-1}$) when incubated in 20°C temperature-controlled room with constant light. Low and high concentration groups refer to the groups used in the Student's t-test, see 2.4 statistics for more information.**

| | Low concentration | | | | | | High concentrations | | | | | |
| | 1 mgL$^{-1}$ | | 10 mgL$^{-1}$ | | 25 mgL$^{-1}$ | | 50 mgL$^{-1}$ | | 75 mg L$^{-1}$ | | 100 mgL$^{-1}$ | |
| Day | Cells ml$^{-1}$ | % Contr. | Cells ml$^{-1}$ | % Contr. | Cells ml$^{-1}$ | % Contr. | Cells ml$^{-1}$ | % Contr. | Cells ml$^{-1}$ | % Contr. | Cells ml$^{-1}$ | % Contr. |
|---|---|---|---|---|---|---|---|---|---|---|---|---|
| 0 | 412 | 96 | 446 | 104 | 246 | 97 | 252 | 99 | 237 | 93 | 231 | 94 |
| 1 | 907 | 101 | 858 | 96 | 712 | 99 | 438 | 61 | 305 | 42 | 271 | 43 |
| 2 | 1107 | 91 | 1110 | 92 | 1530 | 122 | 495 | 40 | 328 | 26 | 313 | 11 |
| 3 | 1167 | 92 | 1197 | 94 | 2117 | 106 | 551 | 28 | 563 | 28 | 396 | 7 |

## 4 Discussion

### 4.1 Dispersal model and experimental design

The current ship dispersal model suggests a dilution rate of 1/1000 over a 2-minute period in the near field of the wake, given a dispersal rate of 500 kg s$^{-1}$. This is consistent with a recent study where the dispersal of $Ca(OH)_2$ from a ship was modeled. The study showed that dilution rates could vary between 710-7100, depending on the diffusion potential of the $Ca(OH)_2$, at a dispersal rate of 100 kg s$^{-1}$, 810 m in the wake behind the ship (Caserini et al., 2021). This distance corresponds to 2 minutes at the modeled ship speed of 25 km h$^{-1}$. Another study from the Cefas Burnham Laboratory, in which maximum (but safe levels of) discharge of industrial waste from ships was sought after, calculated ship discharge dilutions rates of 1/10,000 within 5 minutes was possible (C.Vivian, pers.comm.). Thus, the model in the current paper predicts dilution rates that are within what other model suggests. Still, regarding to models of safe laves of discharge of industry waste, it is important to note that maximum dispersal (discharge) is not the sole criteria for ocean alkalinity enhancement, but rather an intermediate between a high dispersal rate for maximum input and a low dispersal rate to promote maximum dissolution for the alkaline material of choice. For example, in the dispersal model scenario used for designing the experiments in the current study, a 1/10,000 dilution after 1 hour resulted in a final concentration of $Mg(OH)_2$ and $Ca(OH)_2$ of 10 and 12.7 mg/L, respectively. At these concentrations, both alkaline materials are expected to fully dissolve for optimal $CO_2$ uptake while also not resulting in elevated calcium carbonate saturation states leading to "runaway" secondary precipitation of calcium carbonate (e.g., secondary precipitation was observed at $\Omega_{Ar} > 7$ for $Ca(OH)_2$ on the timescale of 4-5 h; Moras et al., 2022 ). Still it cannot be excluded that some uncontrolled $CaCO_3$ precipitation could have occurred at 100 mgL$^{-1}$ of $Mg(OH)_2$ and 127 mgL$^{-1}$ $Ca(OH)_2$ during the initial 1 h of exposure in the present study.

**4.2 Regrowth of *Tetraselmis suecica***

Similar algal densities were observed in both control and $Mg(OH)_2$ treatments at the beginning of the regrowth phase (Day 0, Table 1). This could be related to the short exposure time or to the low solubility of $Mg(OH)_2$; 0.012 g $L^{-1}$ in pure water and around 0.008 g $L^{-1}$ in seawater (Yang et al., 2023). For comparison, the solubility of $Ca(OH)_2$ is 1.73 g $L^{-1}$ at 20-25°C. Accordingly, pH increased during the dispersion phase from approximately 8.0 to 9.5 in the $Mg(OH)_2$ treatment which was lower compared to the expected pH of 12 in $Ca(OH)_2$ (Hartmann et al., 2022). However, pH was similar at the beginning of the regrowth period for both alkaline mineral treatments at ~8.3-8.6 (Fig. 3), giving similar potential regrowth conditions. The similar growth rates observed in controls, $Mg(OH)_2$-added and $Ca(OH)_2$-added treatments (Fig. 2) suggests that the algae previously exposed to 100 g $L^{-1}$ $Ca(OH)_2$ were able to recover during this phase, at least when the algae were incubated in optimal culture conditions which might not be the case in natural oceanic conditions. Taken together, our data indicated high algal mortality in $Ca(OH)_2$ at the high concentrations of 127 g $L^{-1}$ during the first hour after the alkaline mineral discharge from a moving ship, while no such toxic effect was observed when algae were exposed to $Mg(OH)_2$. This emphasizes that the local and temporary biological impact of alkaline mineral discharge in the initial phase of the dispersion, in addition to alkalinity increase capability, needs to be considered when evaluating mCDR strategies. Following this, it is important to keep in mind that in this study the toxicity comparison was based on the criteria that each alkaline mineral should have the same hydroxide content, not taking in account difference in alkalinity enhancement between the alkaline minerals.

**4.3 Growth inhibition test with *Skeletonoma costatum***

The results from the WET tests indicate that no growth inhibition of *S. costatum* was observed for $Mg(OH)_2$ concentrations equal or below to 25 mg $L^{-1}$ (NOEC). This is somewhat in accordance with the simulated dispersion test, showing no growth inhibition of *T. suecica* during the 6 days of regrowth phase in 10 mg $L^{-1}$ magnesium hydroxide. The results from dispersion phase indicate no or low effect of 1 h of exposure with 100 g $L^{-1}$ magnesium hydroxide on *T. suecica*. The WET tests indicated a 50% growth inhibition effect of $Mg(OH)_2$ concentrations ($EC_{50}$) between 82 and 111 mg $L^{-1}$ after 72 h of exposure. This toxicity effect might be explained by the temporary local $CO_2$ limitation impact, limiting the algal growth, due to increasing pH at these high alkaline mineral concentrations. These EC50 values were much higher than $Mg(OH)_2$ solubility of ~ 12.2 mg $L^{-1}$ in pure water (Yang et al., 2023). This raises questions regarding the cause of growth inhibition in the current study. It has been suggested that trace metals, such as Cr, Mo, Ni, Pb in industrial and natural mineral products used as alkaline minerals may impair organism growth (Bach et al., 2019; Hartmann et al., 2022). However, this might not be the case here as the $Mg(OH)_2$ powder used in this study was 97-98% ultrapure with <0.01% Ni or Cr. Further studies are needed to verify and investigate the underlaying mechanism for the growth inhibition of *S. costatum* observed in the current WET tests.

**4.4 Regrowth test with assemblage of ambient algal species**

The same toxicity effect of $Mg(OH)_2$ was observed in the tests performed with local marine algal species; i.e. no significant toxicity effect of $Mg(OH)_2$ concentrations below 25 mg $L^{-1}$ but significant toxicity effect for concentrations above 50 mg $L^{-1}$. *Skeletonoma spp.* was represented in the natural assemblage, as one of the dominant species, while *Skeletonoma costatum* was used in the WET tests. This suggests that the results from the

WET tests using laboratory monoculture are still representative and applicable to similar species growing in natural marine environment. Thus, the results from the natural seawater test demonstrated that toxicity effects observed with $Mg(OH)_2$ on laboratory cultures might be applicable to a wider range of marine algal species.

Thus, both the simulated dispersion scenario, the WET tests and ambient algal tests results suggest that $Mg(OH)_2$ is a suitable alkaline enhancement mineral with respect to minimizing biological impacts on marine microalgae during temporary and local extreme alkaline mineral discharge upon initial phase of the dispersion. While our studies focused on marine microalgae, most other studies focused on the impact of OAE on organisms with calcium carbonate containing parts and therefore sensitive to seawater acidification (Cripps et al.,2013, Fakhraee et al., 2023, Gomes et al., 2016, Renforth and Henderson, 2017). Microalgae play an important role as primary producers and impacts may be reflected in the entire marine ecosystem by affecting higher trophic-level organisms, such as zooplankton and fish (Pauly and Christensen, 1995; Chassot et al., 2010). Accordingly, microalgae are considered a useful and crucial indicator to evaluate the deterioration of environmental quality (Lee et al., 2023). Thus, the current study applying microalgae assays to investigate the effects of $Mg(OH)_2$ suggests a low negative biological impact of $Mg(OH)_2$. However, it is important to keep in mind that these laboratory assays, in addition to proximate the biological impact, are employed because they are relatively fast and cost-effective. Thus, further studies on other functional groups and species are required for ensuring a low impact of the OAE.

**5 Conclusion**

The bioassays based on initial local and temporary discharge simulation from scenario of alkaline mineral dispersion from ship demonstrated that $Mg(OH)_2$ resulted in lower biological impacts on marine microalgae when compared to $Ca(OH)_2$. Further laboratory studies must be completed to include a wider range of biological biodiversity from different trophic levels and on a larger scale, such as in mesocosm studies, prior to field deployment. The observed low negative biological impact of $Mg(OH)_2$ was confirmed by the standardized toxicity test using a more sensitive marine algae species, but also by the tests with a wider range of local ambient marine algal species. Additionally, there are potentially positive biological impacts of OAE, including remediation of ocean acidification conditions by reducing pH and increasing saturation state of calcium carbonate, which were not addressed in this study. Overall, these results indicate that $Mg(OH)_2$ is a suitable mineral for OAE application. Still, it is important to consider that $Mg(OH)_2$ needs to maintain in suspension right below the ocean's surface to be an effective OAE. Thus, in addition to further toxicity assessment of $Mg(OH)_2$ on aquatic environment, techniques for optimization of its dissolution, including injection and distribution methods, in seawater needs to performed.

**6 Data availability**

The raw data are presented in Appendix A for the Tetraselmis test, in Appendix B for the WET tests and in Appendix C for the natural algal assemblage test.

## 7 Author contribution

SD established the study plan, collected all data for data analyses and drafted the first version of this manuscript. EH was involved in statistical analyzes and writing up the manuscript in collaboration with all authors. TN performed the laboratory experiments (both dispersion and regrowth phases) and recorded the biological and chemical analyses results. AK was involved in the quality assurance of the final manuscript.

## 8 Competing interests

NIVA received funding from Negative Emissions Material Inc. (Claymont, USA) to perform the study and from Windward Fund (Washington, USA) for the writing of this publication after results disclosure agreement with Negative Emissions Material Inc. The Windward Fund was founded in response to donors who expressed a desire to be more connected to their peers' work, and to partner with experts in conservation nonprofit management to execute bold initiatives. More info here: https://www.windwardfund.org/about-the-fund/. The authors declare that they have no conflict of interest.

## 9 Acknowledgments

We would like to thank Dr. Evgeniy Yakushev (NIVA) for the development of the BROM model for magnesium hydroxide specific application, Dr. August Tobiesen (NIVA) and Dr. Ana Catarina Almeida (NIVA) for their expertise contribution during test plan and/or experiments execution of bioassays and/or WET tests. This material is based upon work supported by funding from Negative Emissions Material Inc. (Claymont, USA) to perform both the bioassays and the WET tests. NIVA has received funding from Windward Fund (Washington, USA) for the writing of this publication under the Master Services Agreement No. Windward-NOR16-MSA-2023.

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

# Appendix A – Raw data for the *Tetraselmis* bioassay studies

The Table 1 of the manuscript was generated from the raw data presented in Table A1.

**Table A1. Daily averages (n=3) of density of living *Tetraselmis suecica* (cell mL$^{-1}$) during the regrowth phase (Day 0 – Day 6) of the triplicate tests mimicking dispersion of the alkaline minerals Mg(OH)$_2$ or Ca(OH)$_2$ from a ship. Before the regrowth phase, algae were exposed to either 100 g L$^{-1}$ Mg(OH)$_2$ or 127 g L$^{-1}$ Ca(OH)$_2$ (resulting in similar molar concentration of hydroxide in the two alkaline mineral suspensions) for 1 hour. After this, subsamples from each treatment were diluted 10 000 times and algae growth were studied during a 6-day regrowth phase. Each alkaline mineral treatment and corresponding control treatment was assayed in triplicates. Values at day zero corresponds to 1h after dilution and effects of each alkaline mineral was investigated in triplicates.**

| | Density averages (n=3) of living Tetraselmis suecica (cell.mL$^{-1}$) | | | | | | | | | | | |
| | Mg(OH)$_2$ | | | | | | Ca(OH)$_2$ | | | | | |
| | Treated | | | Control | | | Treated | | | Control | | |
| Day/Replicate # | 1 | 2 | 3 | 1 | 2 | 3 | 1 | 2 | 3 | 1 | 2 | 3 |
|---|---|---|---|---|---|---|---|---|---|---|---|---|
| Day 0 | 27 | 30 | 82 | 32 | 31 | 156 | 0 | 0 | 1 | 116 | 152 | 34 |
| Day 1 | 40 | 64 | 84 | 65 | 44 | 159 | 1 | 0 | 1 | 152 | 89 | 39 |
| Day 2 | 72 | 129 | 256 | 115 | 77 | 399 | - | 0 | 3 | - | 361 | 86 |
| Day 3 | 101 | 249 | - | 141 | 125 | - | 6 | 0 | 4 | 1012 | 766 | 110 |
| Day 6 | 1040 | 1533 | 6217 | 1245 | 583 | 4844 | 56 | 1 | 29 | 8275 | 930 | 1230 |

**Table A2. Daily water quality measurements (pH, temperature and salinity) in the treated and control test waters during the 6-day regrowth phase of the triplicate tests (Test 1, Test 2, Test 3) when mimicking dispersion of the alkaline minerals Mg(OH)$_2$ or Ca(OH)$_2$ from a ship.**

| | Mg(OH)$_2$ - Treated water | | | | | | | | | | Mg(OH)2 - Control water | | | | | | | | | |
| | pH | | | Temp.(°C) | | | Salinity (PSU) | | | | pH | | | Temp.(°C) | | | Salinity (PSU) | | |
| days | Test 1 | Test 2 | Test 3 | Test 1 | Test 2 | Test 3 | Test 1 | Test 2 | Test 3 | days | Test 1 | Test 2 | Test 3 | Test 1 | Test 2 | Test 3 | Test 1 | Test 2 | Test 3 |
|---|---|---|---|---|---|---|---|---|---|---|---|---|---|---|---|---|---|---|---|
| 0 | 8.23 | 8.70 | 8.31 | 19.9 | 20.6 | 19.6 | 31.7 | 31.8 | 33.6 | 0 | 7.93 | 8.38 | 7.92 | 18.7 | 19.8 | 18.8 | 31.8 | 31.8 | - |
| 1 | 8.29 | 8.33 | 8.27 | 21.6 | 21.7 | 21.9 | 32.8 | 32.7 | 33.7 | 1 | 8.05 | 8.07 | 8.00 | 21.1 | 21.6 | 22.4 | 32.3 | 32.8 | 33.7 |
| 2 | 8.25 | 8.28 | 8.22 | 21.9 | 21.2 | 22.2 | 33.7 | 32.4 | 34.1 | 2 | 8.12 | 8.13 | 8.08 | 21.3 | 21.3 | 21.8 | 33.3 | 32.7 | 34.0 |
| 3 | 8.20 | 8.26 | - | 21.4 | 21.1 | - | 35.4 | 32.1 | - | 3 | 8.13 | 8.15 | - | 21.2 | 21.1 | - | 34.9 | 32.4 | - |
| 6 | 8.26 | 8.25 | 8.28 | 21.2 | 21.0 | 22.5 | 41.6 | 32.8 | 34.5 | 6 | 8.24 | 8.21 | 8.21 | 21.0 | 21.2 | 22.5 | 40.4 | 33.4 | 34.0 |

| | Ca(OH)$_2$ - Treated water | | | | | | | | | | Ca(OH)$_2$ - Control water | | | | | | | | | |
| | pH | | | Temp.(°C) | | | Salinity (PSU) | | | | pH | | | Temp.(°C) | | | Salinity (PSU) | | |
| days | Test 1 | Test 2 | Test 3 | Test 1 | Test 2 | Test 3 | Test 1 | Test 2 | Test 3 | days | Test 1 | Test 2 | Test 3 | Test 1 | Test 2 | Test 3 | Test 1 | Test 2 | Test 3 |
|---|---|---|---|---|---|---|---|---|---|---|---|---|---|---|---|---|---|---|---|
| 0 | 8.57 | 8.42 | 8.42 | 18.2 | 19.1 | 18.7 | 33.7 | 33.9 | 33.3 | 0 | 7.90 | - | 7.99 | 19.1 | - | 18.6 | 33.9 | - | 33.5 |
| 1 | 8.43 | 8.29 | 8.31 | 21.3 | 23.7 | 22.1 | 33.6 | 35.4 | 33.4 | 1 | 8.08 | 8.13 | 8.09 | 23.3 | 21.0 | 21.4 | 35.5 | 33.6 | 33.7 |
| 2 | 8.37 | - | 8.27 | 21.3 | - | 22.1 | 33.5 | - | 33.4 | 2 | - | 8.16 | 8.14 | - | 21.0 | 21.4 | - | 33.5 | 33.7 |
| 3 | 8.33 | 8.26 | 8.24 | 21.5 | 25.2 | 22 | 33.5 | 33.9 | 32.8 | 3 | 8.23 | 8.17 | 8.14 | 25.1 | 21.0 | 21.4 | 33.7 | 33.5 | 32.7 |
| 6 | 8.24 | 8.36 | 8.24 | 21.4 | 25.4 | 22.1 | 34.5 | 37.3 | 33.9 | 6 | 8.22 | 8.19 | 8.20 | 25.2 | 21.2 | 21.7 | 37.6 | 34.5 | 34.0 |

# Appendix B – Raw data for the WET tests


The Table 3 of the manuscript was generated from the raw data presented in Table B1 and Table B2. The
complete laboratory analysis reports can be provided upon request.

**Table B1. Calibration data for WET Test 1 and for WET Test 2 to correlate the fluorescens measurements to the cell**
**density of _Skeletonoma costatum_. The cell density was determined by fluorescence with SpectraMax iD3 microplate**
**after approximately 72 hours (±2h). The fluorescence measurements were directly correlated to the algal density as a**
**correlation factor ($r^2$) of 1 between the measured fluorescence and the cell density was calculated.**

| WET test 1- calibration data | | WET test 2- calibration data | |
|---|---|---|---|
| **Cell counts** | **Fluorescence** | **Cell counts** | **Fluorescence** |
| 9767 | 21129 | 7722 | 20909 |
| 34407 | 91377 | 28320 | 60447 |
| 105747 | 194737 | 169517 | 267903 |
| 581800 | 1533120 | 543317 | 623790 |


**Table B2. Fluorescens measurements of the control and Mg(OH)$_2$ treatments for WET Test 1 and WET Test 2 after**
**72 hours exposure according to ISO 10253:2016. A total of six concentrations of Mg(OH)$_2$ was tested (1, 10, 25, 50, 75**
**and 100 mg L-1). Each concentration was tested in triplicate, with 6 replicates for each control (one control set with**
**normal ISO 10253 and another control set with modified ISO 10253).**

| | Fluorescense results for WET Test 1-72h | | | | | | | |
|---|---|---|---|---|---|---|---|---|
| | Controls | | Mg(OH)2 concentration in mg.L$^{-1}$ | | | | | |
| Replicate # | Normal control | Modified control | 1 | 10 | 25 | 50 | 75 | 100 |
| 1 | 1741942 | 492151 | 581669 | 854536 | 752064 | 316455 | 227769 | 114436 |
| 2 | 1629608 | 582180 | 593910 | 775861 | 780683 | 334224 | 198120 | 111869 |
| 3 | 1720051 | 332864 | 542791 | 816187 | 705611 | 329265 | 234354 | 113917 |
| 4 | 1885773 | 514530 | | | | | | |
| 5 | 2048400 | 398823 | | | | | | |
| 6 | 1973322 | 481943 | | | | | | |
| | Fluorescence results for WET Test 2-72h | | | | | | | |
| | Controls | | Mg(OH)2 concentration in mg.L$^{-1}$ | | | | | |
| Replicate # | Normal control | Modified control | 1 | 10 | 25 | 50 | 75 | 100 |
| 1 | 2124534 | 640947 | 775797 | 1044538 | 1184687 | 514139 | 168631 | 59714 |
| 2 | 2188199 | 671593 | 713625 | 920976 | 1196415 | 441565 | 212443 | 50273 |
| 3 | 2203985 | 679313 | 713790 | 988564 | 1274252 | 453043 | 170141 | 53626 |
| 4 | 2344184 | 634189 | | | | | | |
| 5 | 2194617 | 445427 | | | | | | |
| 6 | 2209858 | 671270 | | | | | | |




# Appendix C – Raw data for the natural algal assemblage tests

Table 3 of the manuscript was generated from the raw data presented in Table C1 below.

**Table C1. Daily triplicate enumeration of density of living ambient algal cells (cell mL$^{-1}$) with FDA/CMFDA method in**
**Mg(OH)$_2$ treated and control treatments during 3 days of exposure to six different concentrations of Mg(OH)2 (1, 10,**
**25, 50, 75 and 100 mg L-1) when incubated in 20°C temperature-controlled room with constant light. Some of those**
**tests were conducted separately with therefore different control waters. Those tests were carried out on different weeks.**
**Therefore, different control treatments were applied with one control for 1-10 mg/L Mg(OH)$_2$ treatments, one control**
**for 50-75 mg/L Mg(OH)$_2$ treatments and one control for 100 mg/L Mg(OH)$_2$ treatment.**

| | | Densities of living ambient algae (cell.mL$^{-1}$) | | | | | | | | |
| | | Mg(OH)$_2$ Treated (cell.mL$^{-1}$) | | | | | | Control (cell.mL$^{-1}$) | | |
| | | Low concentrations | | | High concentrations | | | for the corresponding treatments with | | |
| | Replicate # | 1 mg.L$^{-1}$ | 10 mg.L$^{-1}$ | 25 mg.L$^{-1}$ | 50 mg.L$^{-1}$ | 75 mg.L$^{-1}$ | 100 mg.L$^{-1}$ | 1-10 mg.L$^{-1}$ | 25-75 mg.L$^{-1}$ | 100 mg.L$^{-1}$ |
|---|---|---|---|---|---|---|---|---|---|---|
| | 1 | 420 | 443 | 220 | 278 | 192 | 212 | 407 | 264 | 240 |
| Day 0 | 2 | 447 | 470 | 254 | 210 | 252 | 250 | 480 | 238 | 276 |
| | 3 | 370 | 423 | 264 | 268 | 266 | 230 | 403 | 258 | 222 |
| | 1 | 955 | 860 | 745 | 400 | 303 | 250 | 875 | 785 | 550 |
| Day 1 | 2 | 895 | 825 | 700 | 450 | 275 | 280 | 910 | 715 | 666 |
| | 3 | 870 | 890 | 690 | 463 | 338 | 282 | 910 | 655 | 662 |
| | 1 | 1040 | 1110 | 1630 | 550 | 338 | 300 | 1340 | 1380 | 2733 |
| Day 2 | 2 | 1120 | 1190 | 1570 | 450 | 330 | 308 | 1000 | 1130 | 3183 |
| | 3 | 1160 | 1030 | 1390 | 485 | 315 | 333 | 1290 | 1240 | 2950 |
| | 1 | 1200 | 1240 | 2000 | 580 | 560 | 377 | 1220 | 1860 | 5925 |
| Day 3 | 2 | 1160 | 1180 | 2280 | 483 | 600 | 400 | 1360 | 2050 | 5425 |
| | 3 | 1140 | 1170 | 2070 | 590 | 530 | 410 | 1240 | 2080 | 4750 |


