# Peer review of "Ocean Alkalinity Enhancement impacts: Regrowth of marine microalgae in alkaline mineral concentrations simulating the initial concentrations after ship-based dispersions"

_Biogeosciences, 2023_

## Author Response (AR1)

*We thank the Editors and reviewers for constructive comments and suggestions for changes. We hope that our replies will make the manuscript publishable in Biogeosciences.*

**Associate editor:**

1) As the model used to calculate TA addition rates from ships and eventual dilutions are subject to a confidentiality agreement, it would be beneficial to compare the results to other published models, putting them into perspective.

*# 1) The following text was added in section 2.1 of the enclosed new revised manuscript (Lines 134-146):*

*A simplified formula for dilution factor based on volume discharge rate, vessel speed, water line depth, and time after disposal was adopted in 1975 by the former International Maritime Consultative Organization (now the International Maritime Organization). Subsequent studies found that the formula underestimated dilution factor (e.g., Byrne et al., 1988). A modeling study similar to the CFD model reported here found that 100 kg s$^{-1}$ and 10 kg s$^{-1}$ Ca(OH)$_2$ addition resulted in 1/166 and 1/52 dilution, respectively, over a ~30 second-period in the near field of the wake zone (Caserini et al., 2021). Despite different ship dimensions and other model inputs including dispersion rate, the dilution rate of 1/1000 over a 2-minute period (this study) was similar for the near field of the wake. Another study from the Cefas Burnham Laboratory, in which maximum (but safe levels of) discharge of industrial waste from ships was sought after, calculated ship discharge dilutions rates of 1/10,000 within 5 minutes was possible (C.Vivian, pers.comm.), however maximum dispersal (discharge) is not the sole criteria for ocean alkalinity enhancement, but rather an intermediate between a high dispersal rate for maximum input and a low dispersal rate to promote maximum dissolution for the alkaline material of choice.*

2) In regard to Reviewer's 1 concern of relatively high TA additions, please make sure to frame your study more clearly, i.e. that it deals with assessing the 'extreme' TA/pH conditions that could be found initially upon mineral additions from ships, prior to dilution.

*#2) We have done the following changes to frame the study to the initial phase of OEA dispersion:*

***Title:*** *Ocean Alkalinity Enhancement impacts: Regrowth of marine microalgae in alkaline mineral concentrations simulating the initial concentrations after ship-based dispersions*

***Abstract:***
*"…, especially the impacts of the initial discharge causing local and temporary extreme alkalinity/pH changes"*

*"…concentrations mimicking the initial high concentrations following a dispersion scenario from a ship"*

***Introduction:***
*"… impact of the initial discharge of high alkaline mineral concentrations upon dispersion causing local and temporary extreme alkalinity/pH changes."*

*"Thus, it is crucial to consider not only the toxicity effect, if any, of the final low alkaline mineral concentration after expected final dilution into ocean, but also the potential initial toxicity effect of the initial hot spot discharge of the alkaline mineral on local organisms. These discharges upon dispersion might be local and temporary but it is important to consider that they would be applied at a global scale. These local and temporary effects will potentially include increased cation levels ($Mg^{2+}$ and $Ca^{2+}$), increased bicarbonate and carbonate ions, pH increase or decrease of dissolved carbon dioxide might cause perturbation hotspots affecting phytoplankton species composition and growth…"*

*"In this study, the biological impact of initial and temporary hot spot discharge of $Mg(OH)_2$ concentrations…"*

***Section 4.2:***
*"This emphasizes that the local and temporarily biological impact of alkaline mineral discharge in the initial phase of the dispersion,…"*

***Section 4.3:***
*"during temporary and local extreme alkaline mineral discharge upon initial phase of the dispersion."*

***Conclusion:***
*"The bioassays based on initial local and temporary discharge simulation from scenario…"*

3) Concerning the comment that $Mg(OH)_2$ is 2.4 times more effective to enhance TA in comparison to $Ca(OH)_2$, that is not correct. If moderate amounts are added of up to a few hundred micro moles/kg, then upon full dissolution each mineral should create the same amount of TA (ignoring impurities). However, at higher additions, beyond the high micro molar range, the calcium carbonate saturation state will eventually be so high that $CaCO_3$ will start precipitating, removing TA. How quickly this will happen depends on several factors, including saturation state or mineral type (compare Moras et al. 2022). There is an additional difference between $Ca(OH)_2$ and $Mg(OH)_2$ which will come to effect at higher TA additions, and that is their different solubilities. With $Mg(OH)_2$, 'only' about 2 mM/kg of TA can be generated before dissolution stops, reaching a pH of about 9.5. In contrast, with $Ca(OH)_2$ about 5 mM/kg can be generated, leading to a pH increase of about 12. At this higher pH precipitation of $CaCO_3$ will be much faster than at pH of 9.5 and also $Mg(OH)_2$ can precipitate, much faster removing TA from a saturated $Ca(OH)_2$ than $Mg(OH)_2$ solution. Hence, at TA additions where secondary precipitation is expected to occur, just looking at two points in time to assess TA generation potential without considering the full kinetic (dissolution as well as precipitation which can even co-occur) is difficult.

REF
Moras et al. 2022: https://doi.org/10.5194/bg-19-3537-2022

*# 3): We agree, and the last sentence of the section 4.2 stating that $Mg(OH)_2$ is 2.4 times more effective than $Ca(OH)_2$ are omitted from the manuscript.*

4) For the new title, I would suggest to slightly change it to: 'The biological impact potential of magnesium hydroxide ....'

*# 4) The tittle of the enclosed revised manuscript has been changed to the following:*

*"Ocean Alkalinity Enhancement impacts: Regrowth of marine microalgae in alkaline mineral concentrations simulating the initial concentrations after ship-based dispersions"*

5) In regard to the comment on the staining protocol to assess dead and alive cells, please make sure that all the necessary clarifications make it into the revised manuscript.

*#5) The corresponding text in section 2.1 is modified, and now states:*

Density of living *Tetraselmis suecica* was determined by using the double staining method with Fluorescein Diacetate (FDA) and 5-chloromethylfluorescein diacetate (CMFDA) (NSF, 2010).For each analysis, a 4 mL subsample was collected and added 4 µl of 10% HCl, bringing the pH back to approximately 8 prior to staining. The sample was then stained by adding 5 mM FDA and 2.5 mM CMFDA stains, as final concentration, and incubated during 10 minutes in the dark. The stained *Tetraselmis suecica* cells were counted in triplicate (3x 1 mL) in a Sedgwick Rafter counting chamber using fluorescence microscope (Leitz Aristoplan, CoolLED pE-300 lite) with 485-530 nm excitation-emission filter combination and 100x magnification. The untreated algal samples without alkaline mineral were used as positive controls. Both *T. suecica* and local diatoms are nearly 100% stainable with these stains according to our 15 years of experience with this method in our local seawater. Samples treated with sodium hydroxide (NaOH) to increase the pH to approximately 14 were used as negative controls. No fluorescence could be observed in the negative controls, indicating an instant kill effect of the algal cells. This study was focusing on the regrowth capability of the algal cells over several days indicated by increasing density of fluorescent cells over time, compared to the control samples. This double staining method FDA/CMFDA is based on the validation work of US Navy Research Laboratory to distinguish between living and dead cells after disinfection by a ballast water treatment (Steinberg et al., 2011). This viability method is the only one recognized by both International Maritime Organization (IMO) and United States Coast Guard (USCG) for approval of ballast water discharge from 70,000 commercial ships at global scale (USCG, 2012, IMO, 2018).

6) Finally, the cell densities and respective survival percentages in the treatment in comparison to the control do not seem to add up. For instance, at 1mg/L on day 3, 1180 cells/ml correspond to 91% while almost twice as many at 25 mg/L are only 109%. Please double-check.

*#6) The reason for the apparent discrepancy is that 1-10, 25-75, 100 mg/l had separate controls. This is now clarified in appendix C (Table C 1), where the cell densities of each control are presented.*

**Table C1. Daily triplicate enumeration of density of living ambient algal cells (cell mL$^{-1}$) with FDA/CMFDA method in Mg(OH)$_2$ treated and control treatments during 3 days of exposure to six different concentrations of Mg(OH)2 (1, 10, 25, 50, 75 and 100 mg L-1) when incubated in 20°C temperature-controlled room with constant light. Some of those tests were conducted separately with therefore different control waters. Those tests were carried out on different weeks. Therefore, different control treatments were applied with one control for 1-10 mg/L Mg(OH)$_2$ treatments, one control for 50-75 mg/L Mg(OH)$_2$ treatments and one control for 100 mg/L Mg(OH)$_2$ treatment.**

| | | Densities of living ambient algae (cell.mL$^{-1}$) | | | | | | | | |
| | | Mg(OH)$_2$ Treated (cell.mL$^{-1}$) | | | | | | Control (cell.mL$^{-1}$) | | |
| | | Low concentrations | | | High concentrations | | | for the corresponding treatments with | | |
| | Replicate # | 1 mg.L$^{-1}$ | 10 mg.L$^{-1}$ | 25 mg.L$^{-1}$ | 50 mg.L$^{-1}$ | 75 mg.L$^{-1}$ | 100 mg.L$^{-1}$ | 1-10 mg.L$^{-1}$ | 25-75 mg.L$^{-1}$ | 100 mg.L$^{-1}$ |
|---|---|---|---|---|---|---|---|---|---|---|
| Day 0 | 1 | 420 | 443 | 220 | 278 | 192 | 212 | 407 | 264 | 240 |
| | 2 | 447 | 470 | 254 | 210 | 252 | 250 | 480 | 238 | 276 |
| | 3 | 370 | 423 | 264 | 268 | 266 | 230 | 403 | 258 | 222 |
| Day 1 | 1 | 955 | 860 | 745 | 400 | 303 | 250 | 875 | 785 | 550 |
| | 2 | 895 | 825 | 700 | 450 | 275 | 280 | 910 | 715 | 666 |
| | 3 | 870 | 890 | 690 | 463 | 338 | 282 | 910 | 655 | 662 |
| Day 2 | 1 | 1040 | 1110 | 1630 | 550 | 338 | 300 | 1340 | 1380 | 2733 |
| | 2 | 1120 | 1190 | 1570 | 450 | 330 | 308 | 1000 | 1130 | 3183 |
| | 3 | 1160 | 1030 | 1390 | 485 | 315 | 333 | 1290 | 1240 | 2950 |
| Day 3 | 1 | 1200 | 1240 | 2000 | 580 | 560 | 377 | 1220 | 1860 | 5925 |
| | 2 | 1160 | 1180 | 2280 | 483 | 600 | 400 | 1360 | 2050 | 5425 |
| | 3 | 1140 | 1170 | 2070 | 590 | 530 | 410 | 1240 | 2080 | 4750 |

*To clarify this, the following text was added at the end of the section 2.3:*

*". Those tests were carried out on different weeks. Therefore, different control treatments applied for 1-10 mg/L Mg(OH)$_2$ treatments, 50-75 mg/L Mg(OH)$_2$ treatments and 100 mg/L Mg(OH)$_2$ treatment (see Appendix C)."*

*Still, some data errors were noticed in the last row of the Table 3 which is now corrected in the enclosed revised manuscript as highlighted in red below:*

*Table 1. Densities of living ambient algal cells (cell mL$^{-1}$) during 3 days of exposure to different concentrations of Mg(OH)$_2$ and their survival in percentage compared to corresponding control treatment without Mg(OH)$_2$ (% Contr.). Cells were incubated in 20°C temperature-controlled room with constant light. Low and high concentration groups refer to the groups used in the Student's t-test, see 2.4 statistics for more information.*

| | Low Mg(OH)$_2$ concentrations | | | | | | High Mg(OH)$_2$ concentrations | | | | | |
| | 1 mg L$^{-1}$ | | 10 mg L$^{-1}$ | | 25 mg L$^{-1}$ | | 50 mg L$^{-1}$ | | 75 mg L$^{-1}$ | | 100 mg L$^{-1}$ | |
| Day | Cells mL$^{-1}$ | % Contr. | Cells mL$^{-1}$ | % Contr. | Cells mL$^{-1}$ | % Contr. | Cells mL$^{-1}$ | % Contr. | Cells mL$^{-1}$ | % Contr. | Cells mL$^{-1}$ | % Contr. |
|---|---|---|---|---|---|---|---|---|---|---|---|---|
| 0 | 412 | 96 | 446 | 104 | 246 | 97 | 252 | 99 | 237 | 93 | 231 | 94 |
| 1 | 907 | 101 | 858 | 96 | 712 | 99 | 438 | 61 | 305 | 42 | 271 | 43 |
| 2 | 1107 | 91 | 1110 | 92 | 1530 | 122 | 495 | 40 | 328 | 26 | 313 | 11 |
| 3 | 1180 | 91 | 1210 | 94 | 2140 | 109 | 531 | 27 | 580 | 30 | 388 | 7 |

Revised and corrected Table 3.

| | Low Mg(OH)$_2$ concentrations | | | | | | High Mg(OH)$_2$ concentrations | | | | | |
| | 1 mg L$^{-1}$ | | 10 mg L$^{-1}$ | | 25 mg L$^{-1}$ | | 50 mg L$^{-1}$ | | 75 mg L$^{-1}$ | | 100 mg L$^{-1}$ | |
| Day | Cells mL$^{-1}$ | % Contr. | Cells mL$^{-1}$ | % Contr. | Cells mL$^{-1}$ | % Contr. | Cells mL$^{-1}$ | % Contr. | Cells mL$^{-1}$ | % Contr. | Cells mL$^{-1}$ | % Contr. |
|---|---|---|---|---|---|---|---|---|---|---|---|---|
| 0 | 412 | 96 | 446 | 104 | 246 | 97 | 252 | 99 | 237 | 93 | 231 | 94 |
| 1 | 907 | 101 | 858 | 96 | 712 | 99 | 438 | 61 | 305 | 42 | 271 | 43 |
| 2 | 1107 | 91 | 1110 | 92 | 1530 | 122 | 495 | 40 | 328 | 26 | 313 | 11 |
| 3 | 1167 | 92 | 1197 | 94 | 2117 | 106 | 551 | 28 | 563 | 28 | 396 | 7 |

**Anonymous Referee 1**

Delacroix et al. test the effect of three potential OAE source materials (Mg(OH)2, Ca(OH)2, NaOH) on phytoplankton. They expose Tetraselmis to extremely alkaline conditions (~3.4 mol/kg added alkalinity, pH 13-14) for one hour and test their survival and regrowth. The extremity of the treatment (and other critical issues in the experimental design) impede drawing meaningful conclusions for OAE. The ecotox tests presented alongside the main experiments are potentially informative for OAE as these may reflect protocols used in permitting procedures in environmental agencies. However, those limited tests with just one species seem too slim to justify a stand-alone publication. Due to these shortcomings (further detailed below) I can unfortunately not recommend publication.

*# 1: According to our 18 years of experience of toxicity risk assessment for the aquatic environment of treated ballast water discharge, phytoplankton species are the most sensitive organisms, in addition to being a critical primary producer for the ocean ecosystem and the most abundant and widely spread organisms in the oceans at global scale, compared to crustacean and fish species for which the toxicity data is indeed also required prior to commercialization permit at global scale. Therefore, we focused only on phytoplankton in this study. Still, we agree in that the next step would be a larger scale mesocosm study including several trophic levels. In fact, this is pointed out in the conclusions in the manuscript. In addition, we want to point out that regarding the effects of MgOH$_2$, the results conducted with a single marine species, were indeed confirmed with an assemblage of different local marine algal species in our study.*

*Regarding the experimental design concerns, we want to point out that the main purpose of our study was to investigate the temporary extreme water quality conditions change at the local contact point of the large amount of dumped alkaline mineral from a moving vessel. This because, virtually all scenarios of OAE require the initial addition of alkaline materials at high concentrations. To mimic this, our experimental was designed to study the short-term toxicity effect on the first phytoplankton exposed to the alkaline mineral discharge in addition to the following long-term exposure effect of a highly diluted alkaline mineral. We hope that the changes in the title to:" Ocean Alkalinity Enhancement impacts: Regrowth of marine microalgae in alkaline mineral concentrations simulating the initial concentrations after ship-based dispersions" together with changes in the abstract and the second paragraph in the introduction makes this clearer.*

*Regarding the comment related to ~3.4 mol/kg added alkalinity and pH 13-14, this might apply to the use of NaOH (this part of the study is now omitted from the manuscript, see further down) as we observed a maximum pH of 9.5 only for example when we used Mg(OH)2. And as mentioned in our article, Hartmann et al. (2022) demonstrated that Mg(OH)2 was 2.4 times more effective in alkalinity enhancement of seawater compared to Ca(OH)2. Therefore, the results of our study might indeed be conservative as our study was aiming in direct toxicity effect comparison when adding a same amount of alkaline mineral only. We agree that further toxicity studies must be conducted when considering equivalent alkalinity increase effect, implying a larger amount of Ca(OH)2 than used in our study and thus probably higher toxicity effect than showed in our study, as we mentioned in our discussion in this manuscript.*

Major comments:

The authors estimated an application rate of 500 kg of alkaline material per second in a ship wake. It is not entirely clear how this number was derived but this is perhaps not the most critical problem. More critically is that this leads to an experimental design where e.g., 141 g of NaOH pellets are added to 1L of Tetraselmis culture for one hour. This seems deadly by design. I am not sure how reliable carbonate chemistry software is under such conditions (i.e., alkalinity = 3.4 mol/kg, DIC = 0.0021 mol/kg) but this translates into a pH of >>13, possibly 14. As such, it is not surprising that even a very robust species like Tetraselmis dies, as the water was essentially sterilized. The fact that Tetraselmis survives the Mg(OH)2 treatment (despite the extremely large amounts added) is most likely due to slower dissolution so that the extreme pH effect cannot fully unfold within an hour of experiment. It could be argued that this is just what happened under such a OAE application and the data is robust (I totally trust the data that all cells died under pH 13-14). However, would anyone consider such application? It would be like doing an ocean acidification experiment under pH 4. Yes, OA reduces pH but pH 4 is still not ocean acidification. Indeed, previous OAE studies have already carefully assessed dosing rates in order to not increase coastal pH beyond a ΔpH of 0.1 on average and pH 11 on the order of seconds assuming a ship wake scenario (He and Tyka, 2023). As such, the way the main experiments are designed will lead to misleading conclusions as what they tested has very little to do with OAE.

The second major critique is the dilution scenario that follows the 1 hour exposure to extremely alkaline conditions. In the oceans, a small volume directly affected by alkaline substances (where some cells may indeed die) would mix with the same water mass and pH would decline to much lower values (e.g. pH 9) within seconds to minutes (see Fig. 7 in He and Tyka, 2023). The not immediately perturbed water body, with which the highly perturbed water mass is quickly mixed, still contains the same unperturbed phytoplankton community. However, in the experiments presented here, deepwater is used because it is essentially free of phytoplankton. As such, the experiments exclude the natural seed population that has not been affected by extremely high pH for an hour. The conclusions drawn from this experiment can therefore not be used to assess an OAE perturbation in a ship wake.

*# Replies to the above major concerns are addressed in the replies to the editor.*

Other comments:

Title: There seems to be a logic issue. Is the impact from OAE or MgOH2? Also, I would refrain from the word "impact" as it implies bias. Influence?

*# Addressed in the answers to the Editor's comments.*

Line 12: Toxicity effect implies by default detrimental outcomes and thus suggests bias. Influence of XXX on YYY?

*# Addressed in the answers to the Editor's comments.*

Line 13: Title mentions MgOH2 but the study seems to address other alkalinity sources.

*#Addressed in the answers to the Editor's comments.*

Line 14: Hydroxyl radical: Isn't that hydroxide ions? The former seems to be used in atmospheric sciences.

 Reply: *# "Hyroxyl radicals" is replaced by "hydroxide ions" in the manuscript.*

First part of introduction: The lengthy introduction on OA seems off-topic and is also not tailored towards the subject of the study, i.e., phytoplankton. The three main negative consequences are also not backed by references and the text seems biased towards considering only negative effects of OA. Pretty much all text until line 51 could be condensed to one sentence or fully deleted, since OA is of little relevance for the research here.

 *# The introduction was shortened by deleting the first part from Line 35 to Line 56.*

Line 53: I would refer to IPCC 2022, I think the relevant working group is either 2 or 3.

*# The text in the Introduction is changed accordingly:*

*"It is widely recognized that reducing the carbon dioxide emissions is not sufficient to accomplish the goals of the Paris agreement of 2015, limiting global warming and ocean acidification (Pathak et al., 2022). «*

*The following literature reference was added in Section 10:*

*Pathak, M., R. Slade, P.R. Shukla, J. Skea, R. Pichs-Madruga, D. Ürge-Vorsatz: Technical Summary. In: Climate Change 2022: Mitigation of Climate Change. Contribution of Working Group III to the Sixth Assessment Report of the Intergovernmental Panel on Climate Change [P.R. Shukla, J. Skea, R. Slade, A. Al Khourdajie, R. van Diemen, D. McCollum, M. Pathak, S. Some, P. Vyas, R. Fradera, M. Belkacemi, A. Hasija, G. Lisboa, S. Luz, J. Malley, (eds.)]. Cambridge University Press, Cambridge, UK and New York, NY, USA. doi: 10.1017/9781009157926.002, 2022.*

Line 55: There are many more, refer to GESAMP 2019.

*# The reference to GESAMP 2019 was added both in the Introduction and section 10 as following:*

*"Many different marine dioxide carbon removal (mCDR) approaches are currently under evaluation (GESAMP, 2019),…"*

*GESAMP: High level review of a wide range of proposed marine geoengineering techniques. (Boyd, P.W. and Vivian, C.M.G., eds.). (IMO/FAO/UNESCO-IOC/UNIDO/WMO/IAEA/UN/UN Environment/UNDP/ISA Joint Group of Experts on the Scientific Aspects of Marine Environmental Protection). Rep. Stud. GESAMP No. 98, 144 p., 2019.*

Line 58: Not all aim to accelerate: Some aim to establish a new C-sink in the earth system that currently does not exist, e.g. seaweed farming.

*# We have rephrased this sentence by adding "some of" in the sentence as following:*

*" In general, the principle of some of these approaches is based on acceleration of the natural process of absorption and long-term storage of the excess atmospheric carbon dioxide by the ocean (Siegel et al., 2021, NASEM, 2021)."*

*Line 60: OAE, when done properly and the generated CO2 deficit is matched with atmospheric CO2, has very little influence on ocean pH. This narrative of OA remediation does therefore make little sense.*

*# We agree with the reviewer and have added the following text:*

*"Hence,when the aquaeous carbon dioxide deficit, generated by the addition of alkaline mineral, returns to the initial equilibrium with atmospheric carbon dioxide, the final pH still remains slightly higher than the initial pH, while calcite (most stable polymorph of calcium carbonate $CaCO_3$) level and aragonite (crystal structure of calcium carbonate) saturation state are elevated. The aragonite saturation state is commonly used to track ocean acidification (Qing-Jiang et al., 2015)."*

Line 66/67: Solubility or dissolution? Have not worked with MgOH2 myself but all data I've seen suggests rather fast dissolution.

*# We meant indeed "solubility" in this sentence. It is indeed the benefit of using an alkaline mineral with low solubility that it is possible to add it in large amount to still get the immediate pH and alkalinity increase effect without inducing precipitation (when added in the right form) and neither toxicity effect on the local marine algae.*

Line 69: Durability is unclear. In principle, all alkalinity should have the same durability.

*# Different alkaline materials might dissolve or react differently, thus resulting in varying degrees of durability of an OAE approach.*

Line 70ff: As mentioned above, I would refrain from "toxicity" in the context as there are likely pros and cons or different organisms. Toxicity implies everything will suffer or die, which is not what current research suggests.

*# A sentence on the potential of positive impacts was added in the conclusion as following: see line 438-444:*

*"Additionally, there are potentially positive biological impacts of OAE, including remediation of ocean acidification conditions by reducing pH and increasing saturation state of calcium carbonate, which were not addressed in this study."*

Line 92: is there a reference that confirms Skeletonema to be more sensitive?

*# References confirming that Skeletonoma is sensitive have been inserted in line 112 and section 10 :*
*"In the second step, toxicity effects of the alkaline minerals were verified by a standardized WET ecotoxicology assay with Skeletonoma costatum, a more sensitive marine algal species (Petersen et al., 2014, Wee et al., 2016), by using the recognized 72 hours growth inhibition test (ISO 10253:2016)."*

*Refences added:*

*International Standard Organization ISO 10253: 2016. Water Quality – Marine algal growth inhibition*

*test with Skeletonema costatum and Phaeodactylum tricornutum. Edition 3. 2016.*

*Petersen K., Heiaas H.H., Tollefsen K.E.: Combined effects of pharmaceuticals, personal care products, biocides and organic contaminants on the growth of Skeletonema pseudocostatum. Aquatic Toxicology. 150:45-54, 2014.*

*Wee J.L., Millie D.F., Nguyen N.K., Patterson J., Cattolico R.A., John D.E., Paul J.H.: Growth and biochemical responses of Skeletonema costatum to petroleum contamination. Journal of Applied Phycology. 28:3317-29, 2016.*

Line 105: Explanation of dilution is unclear. Does that mean a dilutin of 1/1000 after two minutes, 1/7000 after 5 hours, 1/154000 after 10 hours? Please clarify, ideally with an illustrative example.

*# We have rephrased this accordingly as following:*

*"Dilution was observed with an immediate minimum dilution rate of 1/1000 within 2 minutes after injection, followed by an additional minimum dilution rate of 1/7000 during the next 5 hours and a final minimum dilution rate of 1/154000 during the following next 5 hours."*

Line 109: 100g/L appears unrealistically high, which appears to be a critical problem in the study design (see major comment).

*# According to the modelist team's calculations, one of the most economically and realistic scenario was the addition of 500 kg/s of alkaline mineral, resulting in injection of approximately 100 g/L at the speed and ship's length, etc... of this scenario. We choose a high concentration to simulate the worst*

*case to get conservative results. If no or little toxicity effect is observed at this high concentration, we can then safely assume that lower concentrations won't induce any significant toxicity effects either; which might be helpful results for later or further studies.*

Line 174: How was pH calibrated? It seems that best practices for seawater carbonate chemistry measurements (Dickson et al., 2007) were not applied. This may have been necessary but a justification would be required here.

*# The calibration procedure is now described by adding the following sentence in section 2.1:*

*"The three-point calibration method with Hamilton pH-buffer solutions (4, 7 and 10) was used for the calibration of the pH electrode, according to WTW instructions."*

Line 209: Were carbonate chemistry changes through CO2 exchange with the atmosphere considered. These may have altered conditions, especially when cells are grown in beakers.

*# This is addressed by adding the following sentence in Section 2:*

*All experiments were carried out in non-airtight containers to allow ambient $CO_2$ to re-equilibrate with seawater used for the experiments.»*

Line 242: The statistical approach seems to fully counter the experimental design. Why was a gradient established if a range of concentrations were then put together?

*#Because the part of the study with natural assemblage of local algal species was conducted only once, we divided the concentrations in two groups being constituted each by three low or three high concentrations. This approach with three replicates in each group (high and low concentration), allowed us to investigate effects of increasing concentrations statistically. To point this out, the following was added in Line 296-297 in 2.4 Data analysis:*

*" This approach, with three replicates in each group, allowed us to investigate effects of increased MgOH2 concentrations."*

Line 292: The WET test results are somewhat unclear.

*#  The raw data for the WET tests presented in Appendix B.*

Line 321: It is quite obvious that pH was similar at the onset of the regrowth phase because the initial water could diluted. The gradient between treatments is still totally as one would expect it, i.e., highest in NaOH and CaOH2.

*# Here we are unsure how to respond, we want to point out that the parts regarding NaOH are omitted from the study.*

Line 322: The regrowth conditions were extremely different because in some treatments (those that were inadvertently sterilized with NaOH) all cells were dead, as one would expect from a treatment under pH 13-14. (See major comment 2).

*# The part regarding NaOH is omitted from the study, see answer to Major comment No.2.*

Line 362: This statement lacks any evidence, including in the cited references.

*#This sentence was revised accordingly and now sates:*

*"While our studies focused on marine microalgae, most other studies focused on the impact of OAE on organisms with calcium carbonate containing parts and therefore sensitive to seawater acidification (Cripps et al.,2013, Fakhraee et al., 2023, Gomes et al., 2016, Renforth and Henderson, 2017)."*

**Anonymous Referee #2**

In this paper the effect of ocean alkalinity enhancement is tested using different substances simulating discharges from a ship. In the first case Teraselmis sp is exposed to high concentration over a short time, followed by dilution and growth. In addition, further tests with Mg(OH)2 and the diatom Skeletonema costatum and a natural phytoplankton community were done.

All in all, the tests and results are relatively straight forward demonstrating that Mg(OH)2 is compound that affects algal growth the least, but it too will have effects on the survival at high concentrations. As such, it is a nice contribution to the rapidly evolving field of ocean alkalinity enhancement, although not a major breakthrough paper. I noticed the study was funded by a commercial entity and the results were the most favorable for their product, making me wonder if the results had been published if that had not been the case. That is an ethical question to all. I think the transparency about this is good, but would further recommend all of the data to be made fully open in a data repository rather than the 'available upon request' approach that is taken in the present version.

*# The raw data for the Tetraselmis test, WET tests and natural assemblage test were added in Appendix A, Appendix B and Appendix C respectively.*

The conclusion is basically a summary, perhaps apart from mentioning further studies in e.g. mesocosms are needed. Given the results, would you be able to make a comparison with alternatives and perhaps draw up the main bottlenecks (or further study needs) for Mg(OH)2 being the OAE mineral of choice?

*# We have added the following in the conclusion to elucidate challenges that need to be solved to make MgOH$_2$ applicable as an OAE:*

*"Overall, these results indicate that Mg(OH)2 is a suitable mineral for OAE application. Still, it is important to consider that Mg(OH)2 needs to maintain in suspension right below the ocean's surface to be an effective OAE. Thus, in addition to further toxicity assessment of Mg(OH)2 on aquatic environment, techniques for optimization of its dissolution, including injection and distribution methods, in seawater needs to performed. "*

Minor comments

The first paragraph in the intro is not that relevant and could be cut or shortened.

There are a few typographical errors, have a close read through

*# The intro is shortened, and the typos are corrected*

CC1: **'Comment on bg-2023-138'**, Chris Vivian, 22 Aug 2023

Stephanie,

The dilutions you simulated in section 2.1 lines 104-106 are a significant underestimate of what can be achieved in the wake of a vessel. Consequently, your assessment of the safety of discharging alkaline materials from vessels is overly conservative.

I worked at the Cefas Burnham Laboratory for many years and was responsible for advising the Ministry of Agriculture, Fisheries and Food on the licensing of liquid industrial wastes dumped at sea from 1986-1992, including the setting of discharge rates based on toxicity tests. The UK dumped industrial wastes at sea up until 1992 including high strength alkaline wastes. In order to avoid undesirable effects on the marine environment, the wastes were discharged through twin pipes at the stern of converted tankers and calculated dilutions of up to 10,000 times within 5 minutes could be achieved. The dilutions and discharge rates were determined based on toxicity tests.

There are models for the dilution of wastes in the wake of vessels. Paper MEPC III/7 (copy attached) from the 3rd session of IMO MEPC that provided a method for calculation of dilution capacity in a ship's wake. The methodology was derived from field experiments carried out by the United States, Netherlands and Norway, as well as an experiment by the Netherlands using a model in the Netherlands Ship Model Basin. This methodology was used by European countries dumping liquid industrial wastes under the Oslo Convention from the 1970's through to the early 1990's. There are also models for the discharges from exhaust gas cleaning systems on ships (using the MAMPEC-BW model originally developed for ballast water discharges - https://www.deltares.nl/en/software/mampec/). Also, see the attached paper by Caserini et al. (2021).

In the UK's case, the MEPC paper methodology was used to ensure that dilution of the waste was such that the 96-hour LC50 concentration derived from toxicity testing of the wastes was achieved within 5 minutes of discharge. In practice, a number of field experiments carried out by the UK found that the methodology in MEPC III/7 underestimated the dilution achieved in practice in these and other experiments (see attached papers). Speed of vessels, waterline length and discharge rate were the most important factors.

Chris Vivian.

*Chris.vivian2@btinternet.com*

*# We thank the reviewer for pointing out that our study design is rather conservative. The dilution factors suggested by the model in our manuscript were indeed based on minimum estimations. Our model is related with other dispersion models by adding following to section 2.1:*

*"A simplified formula for dilution factor based on volume discharge rate, vessel speed, water line depth, and time after disposal was adopted in 1975 by the former International Maritime Consultative Organization (now the International Maritime Organization). Subsequent studies found that the formula underestimated dilution factor (e.g., Byrne et al ., 1988). A modeling study similar to the CFD model reported here found that 100 kg s$^{-1}$ and 10 kg s$^{-1}$ Ca(OH)$_2$ addition resulted in 1/166 and 1/52 dilution, respectively, over a ~30 second period in the near field of the wake zone (Caserini et al., 2021). Despite different ship dimensions and other model inputs including dispersion rate, the dilution rate of 1/1000 over a 2 minute period (this study) was similar for the near field of the wake. Another study from the Cefas Burnham Laboratory, in which maximum (but safe levels of) discharge of industrial waste from ships was sought after, calculated ship discharge dilutions rates of 1/10,000 within 5 minutes was possible (C.Vivian, pers.comm.), however maximum dispersal (discharge) is not the sole criteria for ocean alkalinity enhancement, but rather an intermediate between a high dispersal rate for maximum input and a low dispersal rate to promote maximum dissolution for the alkaline material of choice."*

**CC2: 'Reply on CC1', Michael Tyka, 30 Aug 2023**
I'll add that it would be quite interesting for the reader to directly see the results of the simulations carried out in section 2.1 (in particular the dilution-vs-time curves) directly compared with the empirical formulas used by IMCO, Chou et al and others. See for example - section 3.3 in https://bg.copernicus.org/articles/20/27/2023/ and Box 2 in https://agupubs.onlinelibrary.wiley.com/doi/full/10.1002/2016RG000533 and also the above-mentioned Caserini et al. (2021). Can we gain any insight from your simulations about the accuracy of these different formulas (which already deviate from each other quite significantly) ?

*# Our model is now related with other dispersion models by adding following to section 2.1:*

*"A simplified formula for dilution factor based on volume discharge rate, vessel speed, water line depth, and time after disposal was adopted in 1975 by the former International Maritime Consultative Organization (now the International Maritime Organization). Subsequent studies found that the formula underestimated dilution factor (e.g., Byrne et al ., 1988). A modeling study similar to the CFD model reported here found that 100 kg s$^{-1}$ and 10 kg s$^{-1}$ Ca(OH)$_2$ addition resulted in 1/166 and 1/52 dilution, respectively, over a ~30 second period in the near field of the wake zone (Caserini et al., 2021). Despite different ship dimensions and other model inputs including dispersion rate, the dilution rate of 1/1000 over a 2 minute period (this study) was similar for the near field of the wake. Another study from the Cefas Burnham Laboratory, in which maximum (but safe levels of) discharge of industrial waste from ships was sought after, calculated ship discharge dilutions rates of 1/10,000 within 5 minutes was possible (C.Vivian, pers.comm.), however maximum dispersal (discharge) is not the sole criteria for ocean alkalinity enhancement, but rather an intermediate between a high dispersal rate for maximum input and a low dispersal rate to promote maximum dissolution for the alkaline material of choice."*

**CC3: Mackenzie Burke, 29 Sep 2023   reply**

Delacroix et al. present the outcomes of a set of experiments from which they argue that Mg(OH)$_2$ is the ideal compound choice for OAE over NaOH and Ca(OH)$_2$. We question these conclusions on two

grounds. First, the treatments are not equivalent because of differences in solubility in the alkaline materials used; second, the staining method used is likely to be an unreliable way to assess the impact of alkalization.

The hydroxides used in the study were added at equivalent molar concentrations. However, there are significant differences in solubility. We calculate the concentrations of added hydroxide ions in the initial treatment phase as 0.311, 25.1, and 353 mmol $L^{-1}$ for $Mg(OH)_2$, $Ca(OH)_2$, and NaOH, respectively, using Ksp values collated by UofMass (https://owl.oit.umass.edu/departments/Chemistry/appendix/ksp.html) for the first two and assuming NaOH was fully dissociated. This would give a range of exposures that vary by 3 orders of magnitude as opposed to being equivalent.

*#We are aware that alkaline metals have different solubilities, resulting in different temporal concentrations of hydroxide ions. However, as we state in the introduction; "there will be considerable local/regional effects of dispersion of alkaline minerals. Increased cation levels (Mg2+ and Ca2+), increased bicarbonate and carbonate ions, temporary local pH increase, or temporary local decrease of dissolved carbon dioxide might cause perturbation hotspots". Thus, in the chosen scenario, also including tonnage capacity and operating costs of a ship, we do not think it is correct to compensate for differences in solubility in the initial concentration. This is because a lower initial concentration will affect the cost efficiency of the dispersion of the alkaline metals. The study design was based on some degree of realism related to dispersion of alkaline materials. Furthermore, as mentioned in the discussion, Hartmann et al. (2022) demonstrated that double the amount of Ca(OH)_2 compared to Mg(OH)_2 had to be dispersed onto the ocean's surface to reach the same alkalinity increase effect. Considering this, the toxicity results of Ca(OH)_2 in our study is conservative as double amount of Ca(OH)_2 than Mg(OH)_2 should actually have been added.*

The ability of FDA+CMFDA as a means for discriminating between living and dead phytoplankton was tested by MacIntyre and Cullen (2016). It was effective in less than half of the 24 species tested when living cells were compared to killed ones, as recommended in the NSF protocols cited by the authors. An important difference between the approach used by MacIntyre and Cullen and the NSF protocols is the use of flow cytometry rather than microscopy. Cytometry has the advantage of providing a very high number of objective and quantitative estimates of fluorescence intensity. None of the test strains used in this study (Tetraselmis suecica, Phaeodactylum tricornutum, Skeletonema marinoi [costatum]) were tested by MacIntyre and Cullen. However, using the same protocols (flow cytometric analysis of cells that were in balanced growth, i.e., all alive, vs heat-killed), we tested Tetraselmis suecica (CCMP906), obtained from the NCMA (https://ncma.bigelow.org/). Both the living and dead cells had high fluorescein fluorescence compared to cells that had not been exposed to the dyes. However, there was high overlap between the frequency distributions of the staining intensity in live and dead cells (Figure A, B). If the live cells were classified using the 95th percentile of the distribution of heat-killed controls, 60% are not different (i.e., they were false negatives — living cells classified as dead) and only 40% were accurately classified as living.

A general failure of the combined stains to discriminate between living and dead cells would not account for the differences between the treatments presented in Tables 1 and 2, but the pH-dependence of the stains might. Alkalization appears to cause partial hydrolysis of FDA in cell-free seawater, evident as an increase in fluorescence due to the release of no-longer-quenched fluorescein (Figure C). Increased pH-dependent lysis would reduce the concentration of the substrate available for staining the cells. A comparison of staining intensity in control and alkalized Diachronema (formerly Pavlova) lutheri (CCMP1325) in balanced growth is shown in Figure D, E. Classification of the culture in the growth medium (D) and in the same cultures alkalized with a

final concentration of 1 mmolar NaOH (E), shows that most living cells have much higher fluorescence than the heat-killed controls. However, even though there is a reduction in the number of cells mis-classified as dead with alkalization (5% vs 13%), the median staining intensity in the alkalized cells is 4x lower than in the controls. In the absence of the appropriate killed controls, these live cells would largely be classified as dead.

Uncertainties over the validity of the means used to classify cells as living or dead could be addressed with additional testing and documentation.

Mackenzie Burke, Jessica Oberlander, Mikaela Ermanovics, Marie Egert, Hugh MacIntyre (Dalhousie University)

MacIntyre HL, Cullen JJ (2016) Classification of phytoplankton cells as live or dead using the vital stains fluorescein diacetate and 5-chloromethylfluorescein diacetate. J Phycol 52 (4):572-589. doi:10.1111/jpy.12415

*#The text in section 2.1 was modified to verify the validity of the method:*

*"Density of living Tetraselmis suecica was determined by using the double staining method with Fluorescein Diacetate (FDA) and 5-chloromethylfluorescein diacetate (CMFDA) (NSF, 2010).For each analysis, a 4 mL subsample was collected and added 4 μl of 10% HCl, bringing the pH back to approximately 8 prior to staining. The sample was then stained by adding 5 mM FDA and 2.5 mM CMFDA stains, as final concentration, and incubated during 10 minutes in the dark. The stained Tetraselmis suecica cells were counted in triplicate (3x 1 mL) in a Sedgwick Rafter counting chamber using fluorescence microscope (Leitz Aristoplan, CoolLED pE-300 lite) with 485-530 nm excitation-emission filter combination and 100x magnification. The untreated algal samples without alkaline mineral were used as positive controls. Both T. suecica and local diatoms are nearly 100% stainable with these stains according to our 15 years of experience with this method in our local seawater. Samples treated with sodium hydroxide (NaOH) to increase the pH to approximately 14 were used as negative controls. No fluorescence could be observed in the negative controls, indicating an instant kill effect of the algal cells. This study was focusing on the regrowth capability of the algal cells over several days indicated by increasing density of fluorescent cells over time, compared to the control samples. This double staining method FDA/CMFDA is based on the validation work of US Navy Research Laboratory to distinguish between living and dead cells after disinfection by a ballast water treatment (Steinberg et al., 2011). This viability method is the only one recognized by both International Maritime Organization (IMO) and United States Coast Guard (USCG) for approval of ballast water discharge from 70,000 commercial ships at global scale (USCG, 2012, IMO, 2018)."*

---

## Referee Report (RR1)

This study reports on the analysis of the effects of the exposure of phytoplankton to ocean liming by using brucite  (Mg(OH)$_2$ and Ca(OH)$_{2)}$ mimicking the initial concentrations following a dispersion scenario from a ship. Three experiments were done: 1) exposure of the marine chlorophyte *Tetraselmis suecica* to simulated dispersion of the minerals from an allegedly moving ship; 2) growth rate inhibition of *Skeletonema costatum* exposed to Mg(OH) by using a marine algal growth inhibition test (called Whole Effluent Toxicity test i.e. WET)*;* 3) exposure of a natural plankton community from the Oslo fjord to increasing concentrations Mg(OH)$_2$.

The authors conclude that Mg(OH$_{)2}$ is a suitable mineral for OAE application. This was supported by: 1) high *T. suecica* mortality at high  Ca(OH)$_2$ concentrations during the first hour of a supposedly discharge from a moving ship *vs.* no mortality to Mg(OH)$_2$ discharge. For this,  green cell viability fluorescence stains were used; 2) Cell numbers of *S. costatum* assessed by fluorescence (666nm? ) showed apparently no differences between  LOEC and NOEC  for Mg(OH$_{)2}$  while  EC50 was 2 and 3-fold higher concentration regarding to LOEC and NOEC respectively; 3) there was a significant difference in algal survival between the low and high Mg(OH$_{)2}$ concentrations three days of exposure of diatoms, dinoflagellates and other unspecified organisms from a natural phytoplankton community.

The outcome of these experiments seems to be that bioassays based on initial local and temporary discharge simulation from alkaline mineral dispersion from ships, demonstrated that Mg(OH)$_2$ resulted in lower biological impacts on marine microalgae when compared to Ca(OH)$_2$.

Unfortunately,  I am afraid I cannot support publication in BG since in my opinion, there are important flaws regarding the experimental design, data interpretation and overstated conclusions, that prevent this work to meet the required quality to be published in BG.

The arguments in which my recommendation is based are the following:

**MAJOR COMENTS**

1. One major concern is that authors use the computational models initially , to set-up the background and aims for the subsequent experiments. However, I see there is a huge gap in between the computational models output and what it was simulated in the lab experiments. Especially, regarding the carbonate system, speciation, pH and precipitation.

Has not pH increase undesirably fast? And so, it is possible that uncontrolled CaCO3 precipitation could lower the CO2 sequestration efficiency of the approach? Theoretically Ca(OH)2, and (Mg(OH)2) should dissolve rapidly in the ocean surface , but I am not sure this the case. I have the impression that  the ratios dissolution /precipitation were not controlled to check for reactivity and spontaneous

precipitation in seawater. In short, were these hydroxides well dissolved? In addition, and regarding the commercially sourced material, was the carbonate content measured before the experiment to check for carbonation? I mean, how can you limit carbonation being present within the hydroxides?

I have serious difficulties concealing the results obtained from de lab experiments with the just commented above. Could your results be generalised, and compared to the simulated computer models and extrapolated to general conclusions that can have relevance for the application of OAE in the real-world? I am not sure…

2. I have the impression that the use of the double staining method with Fluorescein Diacetate (FDA) and 5-chloromethylfluorescein diacetate (CMFDA) have been used lacking rigour (just my opinion and experience).

It is important to note that fluorescent stains must be validated in each different experiment before use by analysing the optimal dye loading concentration and loading kinetics for each specific monoculture and /or phytoplankton community, to avoid sub-optimal fluorescence or saturated fluorescence reaching the laser detectors of flow cytometers or epifluorescence microscopes. This essential step seems to have been omitted previously to the start of the experiment. The concentration used by the authors in this experiment is 2 orders of magnitude higher (2.5 mM final concertation) than working concentrations widely used in different marine coastal and open oceanic waters , as well as in lab cultures by well stablished protocols and SOPs (please check references at the end of the review). Attending to this, the lack of mortality can be an artefact due to the spill over of green fluorescence due to excess oversaturated signal, which commonly occurs when the fluorescent dye concentration has not been customised for every cell type.

For example, for CMFDA long-term staining (more than about 3 days) or the use of rapidly dividing cells, 5–25 µM dye is required. Less dye (0.5–5 µM) is usually needed for shorter experiments, such as viability assays in cultures and about 20 µM in natural populations (always final concentrations) as it is the case in this study. To maintain normal cellular physiology and reduce potential artefacts, as already mentioned, the dye concentration must be kept as low as possible. The effects of overloading may not be apparent, hence, to check for this, a cell death stain must be used in combination in the same set of aliquots containing the cells aim of study (please check references at the end of the review).

In addition, the CMFDA fluorescent probe is well retained in living cells through several generations. The probe is transferred to daughter cells but are not transferred to adjacent cells in a population. Cells loaded with the CMFDA fluorescent probes display fluorescence for at least 72 hours and exhibit ideal tracking dye properties—they are stable, nontoxic at working concentrations well retained in cells, and brightly fluorescent at physiological pH. Therefore, assessing cell viability with this fluorescent probe is not entirely accurate since daughter cells can be metabolically non-viable and yet, show green fluorescence. In this case, cell viability is overestimated (please check references at the end of the review).

Regarding FDA I have similar concerns except for the transfer of dye to the daughter cells that is not the case with FDA.

3. I must also comment that I miss the detailed staining protocol for cells and fluorescent microscopy quantification. How were cells harvested: by centrifugation and aspirate the supernatant? By filtration? Were they resuspended in pre-warmed or RT working Solution ? Gently or vortexed? For how long were cells incubated with the dyes? Were cells centrifuged to remove the excess dye working solution? Was culture media added and the labelled cell dispensed onto slide or into a chamber-wells for imaging? For how long were they imaged using the appropriate emission and excitation filters under the scope? The point being, if someone is to reproduce your experiment, not sure that would be possible with the insufficient information provided…

The kinetics and loading curves set-up for the optimal dye concentration and time of incubation should be provided in supplemental material, or at least mention in the text that they are available to reviewers in case they would like to check on them (as it is my case for example)

4. More details on negative and positive controls choice would be desirable. Were general procedural negative controls were done?

5. I was wondering why in the case of *Tetraselmis* a flow cytometer was not used… this would have most likely had produced more accurate results as compared to microscope observations. Regarding the scope is not clear how the % of viable cells was calculated nor how representative your sample was. What was N? how many fields of view (FOVs) were counted per slide or well? In each of the independent cultures? If only one slide/independent culture counted, seems not enough to me. Not clear either which statistical analyses has been carried out for this? Epifluorescence (or any microscopically quantification) can lead to artefactual data unless N is large enough (not sure this is the case, and clarification is needed), or other intercalibrating method is used to contrast with numbers, such as flow cytometry.

6. Another question that is confusing to me is the lack of standardised methods for measuring cell performance. It is not clear enough why fluorescent probes were used with *Tetraselmis*, but not for Skeletonema nor for the natural community. Moreover, for Skeletonema fluorescence, authors do not specify which fluorescence was measured? Red 666nm? In which device was this measured? Green with probes? Again, in which device? Along the same line, the rationale for the natural community analysis it is not well understood, nor how the % of viable cells was also calculated. Table 3 and 4 are difficult to understand because it cannot be discriminated to which functional group each % belong, therefore, not sure I see the point for this.

I'd like to point out that the methodology description does not suffice to understand how this experiment was performed. The ms. does not have an easy thread-line to be followed. Could it perhaps be better organised?

7. The statistical approach used, does not seem appropriate. First, does data distribution meet the requirements for parametrical tests? Assuming so,  T-tests do not capture the variability of the system you might have. Hence, most likely, significant differences are not well resolved, questioning the results. I would suggest that a one-way ANOVA would be adequate since you have 5 levels of concentrations.  Also, in those cases in which time is a continuous variable and not an end-point variable, the right approach could be a split-plot ANOVA in which the fixed factor would be the concentration and the repeated measures factor would be time.

8. In my opinion the discussion is shallow. It does not get deep insight on explaining the data, nor debating them. Not enough quality for a discussion I'm afraid. The same applies to the conclusions, which are merely descriptive.

**MINOR COMMENTS**

Ln. 139. The statement "were repeated three times for each alkaline mineral" to what this exactly refers? Repeated when? How would your N then vary?

Ln. 214. "A preliminary study was made to verify the microalgal growth in this modified media". Why is this not shown?

Ln.221. What do you envision would have occurred if instead of applying the minerals to log-phase cells , they would have been added in lag-phase (in which many cells are in natural conditions at sea)? Would have you expected any growth? This particularity of the growth curve should have been tested too. The outcome can be surprising and non-acclimation i.e. death, shall be considered as a very likely possibility, questioning the application of these chemicals in the ocean.

Ln. 236. "25 L grab-sample from the surface water of Oslofjord
was directly used for the test or a 2 L subsample was mixed to 2 L of 60 m deep seawater from Oslofjord". You are mixing surface cells acclimated at a probable high irradiance with water at 60m depth and submitting them to a few umols  photons m-2s-1. This might for sure have consequences in cell viability, just because metabolism can change due to the new light scenario. Metabolic activity is reflected by viability cell stains since they are dependent on esterases inside the cell.

Ln 282. Perhaps this table would be best in supplementary material?

References:

1. McIntyre, H. L. & Cullen, J. J. 2016. Classification of phytoplankton cells as live or dead using the vital stains fluorescein diacetate and 5-chloromethylfluorescein diacetate.J. Phycol. 52, 572–589.
2. Peperzak, L. & Brussaard, C. P. D. 2011. Flow cytometric applicability of fluorescent vitality probes on phytoplankton. J. Phy- col. 47:692–702.

3.  Segovia, M., & Berges, J. A. (2009). Inhibition of caspase-like activities prevents the appearance of reactive oxygen species and dark-induced apoptosis in the unicellular chlorophyte Dunaliella tertiolecta. Journal of Phycology, 45(5), 1116–1126.
4.  Segovia, M., Lorenzo, M. R., Maldonado, M. T., Larsen, A., Berger, S. A., Tsagaraki, T. M., & Egge, J. K. (2017). Iron availability modulates the effects of future $CO_2$ levels within the marine planktonic food web. Marine Ecology Progress Series, 565, 17-33.
5.  Sobrino, C., Segovia, M., Neale, P. J., Mercado, J. M., García-Gómez, C., Kulk, G., & Ruan, Z. (2014). Effect of $CO_2$, nutrients and light on coastal plankton: Physiological responses. Aquatic Biology, 22, 77-93
6.  Steinberg, M. K., Lemieux, E. J. & Drake, L. A. 2011. Determining the viability of marine protists using a combination of vital, fluorescent stains. Mar. Biol. 158:1431–7.
7.  Veldhuis, M. J. W., Cucci, T. L. & Sieracki, M. E. 1997. Cellular DNA content of marine phytoplankton using two new fluorochromes: taxonomic and ecological implications. J. Phycol. 33:527–41.
8.  Veldhuis, M. J. W., Kraay, G. W. & Timmermans, K. R. 2001. Cell death in phytoplankton: correlation between changes in membrane permeability, photosynthetic activity, pigmentation and growth. Eur. J. Phycol. 36:167–77.

---

## Author Response (AR2)

Response to 2.nd round review:

We thank the reviewer and the editor for valuable comments and hope that our responses to the concerns will make the manuscript publishable in biogeosciences.

MAJOR COMENTS 1. One major concern is that authors use the computational models initially, to setup the background and aims for the subsequent experiments. However, I see there is a huge gap in between the computational models output and what it was simulated in the lab experiments.

**In the first review round it was pointed out that we needed to compare our model for dispersal with other models. In the latest version this was included in the M&M. This part is now re-written and moved to the discussion (4.1. Dispersal model and experimental design). We hope that this will make the part of the M&M linking the dispersion model to the experimental condition clearer.**

Especially, regarding the carbonate system, speciation, pH and precipitation. Has not pH increase undesirably fast? And so, it is possible that uncontrolled CaCO3 precipitation could lower the CO2 sequestration efficiency of the approach? Theoretically Ca(OH)2, and (Mg(OH)2) should dissolve rapidly in the ocean surface , but I am not sure this the case. I have the impression that the ratios dissolution /precipitation were not controlled to check for reactivity and spontaneous precipitation in seawater. In short, were these hydroxides well dissolved? In addition, and regarding the commercially sourced material, was the carbonate content measured before the experiment to check for carbonation? I mean, how can you limit carbonation being present within the hydroxides? I have serious difficulties concealing the results obtained from de lab experiments with the just commented above. Could your results be generalised, and compared to the simulated computer models and extrapolated to general conclusions that can have relevance for the application of OAE in the real-world? I am not sure...

**At 100 mg/L of  $Mg(OH)_2$  and  $Ca(OH)_2$ , there is the possibility that some uncontrolled  $CaCO_3$ precipitation could have occurred. But following the 1 hr "dispersal phase", the 10,000x dilution resulted in 10 mg/L  $Mg(OH)_2$  and 12.7 mg/L  $Ca(OH)_2$  which would result in omega-aragonite and omega-calcite saturation states that would not result in uncontrolled  $CaCO_3$  precipitation. However, carbonate chemistry was not comprehensively measured in these experiments. Neither was the carbonate content of the commercially sourced materials. The dilutions, however, were designed after model dispersal results, so similar carbonate chemistry conditions should be expected in a real-world OAE deployment from a ship into the open ocean.**

*#* To discuss eventual precipitation the the following are added to the discussion:

"For example, in the dispersal model scenario used for designing the experiments in current study, a 1/10,000 dilution after 1 hour results in a final concentration of Mg(OH)2 and Ca(OH)2 of 10 and 12.7 mg/L, respectively. At these concentrations, both alkaline materials are expected to fully dissolve for optimal CO2 uptake while also not resulting in elevated calcium carbonate saturation states leading to "runaway" secondary precipitation of calcium carbonate (e.g., secondary precipitation was observed at  $\Omega Ar > 7$  for Ca(OH)2 on the timescale of 4-5 h; Moras et al., 2022 ). Still it cannot be excluded that some uncontrolled CaCO3 precipitation could have occurred at 100 mgL-1 of Mg(OH)2 and 127 mgL-1 Ca(OH)2 during the initial 1 h of exposure in the present study."

2. I have the impression that the use of the double staining method with Fluorescein Diacetate (FDA) and 5-chloromethylfluorescein diacetate (CMFDA) have been used lacking rigour (just my opinion and experience). It is important to note that fluorescent stains must be validated in each different experiment before use by analyzing the optimal dye loading concentration and loading kinetics for

each specific monoculture and /or phytoplankton community, to avoid sub-optimal fluorescence or saturated fluorescence reaching the laser detectors of flow cytometers or epifluorescence microscopes. This essential step seems to have been omitted previously to the start of the experiment. The concentration used by the authors in this experiment is 2 orders of magnitude higher (2.5 mM final concertation) than working concentrations widely used in different marine coastal and open oceanic waters, as well as in lab cultures by well stablished protocols and SOPs (please check references at the end of the review). Attending to this, the lack of mortality can be an artefact due to the spill over of green fluorescence due to excess oversaturated signal, which commonly occurs when the fluorescent dye concentration has not been customised for every cell type. For example, for CMFDA long-term staining (more than about 3 days) or the use of rapidly dividing cells,  $5-25 \mu$ M dye is required. Less dye (0.5–5  $\mu$ M) is usually needed for shorter experiments, such as viability assays in cultures and about 20  $\mu$ M in natural populations (always final concentrations) as it is the case in this study. To maintain normal cellular physiology and reduce potential artefacts, as already mentioned, the dye concentration must be kept as low as possible. The effects of overloading may not be apparent, hence, to check for this, a cell death stain must be used in combination in the same set of aliguots containing the cells aim of study (please check references at the end of the review).

**We thank the reviewer for making us aware about a typo regarding dye concentrations. It should be  $\mu$ M instead of mM. This is changed in the manuscript. Regarding to validation, the FDA/CMFDA method has been used by us for viability staining of Tetraselmis sp. since 2016 when it was compulsory method for USCG testing of BWMS. We perform in-house reproducibility testing of operators to ensure performance quality acceptance.**

In addition, the CMFDA fluorescent probe is well retained in living cells through several generations. The probe is transferred to daughter cells but are not transferred to adjacent cells in a population. Cells loaded with the CMFDA fluorescent probes display fluorescence for at least 72 hours and exhibit ideal tracking dye properties—they are stable, nontoxic at working concentrations well retained in cells, and brightly fluorescent at physiological pH. Therefore, assessing cell viability with this fluorescent probe is not entirely accurate since daughter cells can be metabolically non-viable and yet, show green fluorescence. In this case, cell viability is overestimated (please check references at the end of the review). Regarding FDA I have similar concerns except for the transfer of dye to the daughter cells that is not the case with FDA.

**The samples were stained 10 minutes prior to counting. Counting was performed within a maximum of 45 minutes. The staining of dead daughter cell would not occur within this time frame. In addition, elevated background green fluorescence would occur long before any dead cell showed elevated fluorescence.**

3. I must also comment that I miss the detailed staining protocol for cells and fluorescent microscopy quantification. How were cells harvested: by centrifugation and aspirate the supernatant? By filtration? Were they resuspended in pre-warmed or RT working Solution? Gently or vortexed? For how long were cells incubated with the dyes? Were cells centrifuged to remove the excess dye working solution? Was culture media added and the labelled cell dispensed onto slide or into a chamber-wells for imaging? For how long were they imaged using the appropriate emission and excitation filters under the scope? The point being, if someone is to reproduce your experiment, not sure that would be possible with the insufficient information provided... The kinetics and loading curves set-up for the

optimal dye concentration and time of incubation should be provided in supplemental material, or at least mention in the text that they are available to reviewers in case they would like to check on them (as it is my case for example).

**The paragraph regarding staining has been rewritten and now states: "The density of living Tetraselmis suecica was determined using the double staining method with fluorescein diacetate (FDA) and 5-chloromethylfluorescein diacetate (CMFDA) (NSF, 2010). This double staining method, FDA/CMFDA, is based on the validation work of the US Navy Research Laboratory to distinguish between living and dead cells after disinfection by a ballast water treatment (Steinberg et al., 2011). This viability method is the only one recognized by both the International Maritime Organization (IMO) and the United States Coast Guard (USCG) for approval of ballast water discharge from 70,000 commercial ships at a global scale (USCG, 2012, IMO, 2018).**

The following staining protocol was used: A 2.5 mM CMFDA stock solution was prepared by dissolving 1 mg of CMFDA in 0.86 ml DMSO (Dimethyslsulphoxide). It was then divided into 50  $\mu$ l batches and stored at -20 °C. The 5 mM FDA stock solution was prepared by dissolving 10 mg FDA in 4.8 ml DMSO. The FDA stock solution was divided into 100  $\mu$ l batches and stored at -20 °C. For each analysis, a 4 ml subsample was collected and 4  $\mu$ l of 10% HCl was added, bringing the pH back to approximately 8 prior to staining. 4  $\mu$ l of each stock solution was added to each subsample, resulting in final concentrations of 2.5  $\mu$ M CMFDA and 5 $\mu$ M FDA. The subsamples were then incubated in darkness for 10 minutes, after which they were loaded into 1 ml Sedgewick-Rafter counting chambers etched with 1-mm 2 grids, with 1000 fields of view (FOV). Chambers were examined at 100x magnification using compound epifluorescent microscopes with standard blue light excitation (480 nm) and green bandpass emission (530 nm) filters. Furthermore, FOVs were counted until a minimum of 100 viable cells was observed in each camber. Cells in 3 replicate chambers were counted for each sample. For samples with <100 cells/ml, cells in up to six chambers were counted. Samples were counted within a 45-minute period after incubation Cells numbers in sample were set to zero If no cells were observed in six chambers. Viable cells in samples (%) vere calculated as 100 \* cell number in treated sample/cell numbers is control sample (without alkaline at each specific day, in this case). Average cell numbers were presented in tables in the MS."

4. More details on negative and positive controls choice would be desirable. Were general procedural negative controls were done? Is this the NaOH teratement

**Negative controls were unamended seawater which are described at the end of section 2.1, 3rd paragraph as well as in Fig. 1. Positive controls were the NaOH treatments described in section 2.1, 8th paragraph. The negative controls were with no additions and represent phytoplankton growth/physiology under normal conditions. The positive controls received an extreme high pH treatment that is intended to have an extreme negative impact on phytoplankton growth/physiology.**

5. I was wondering why in the case of Tetraselmis a flow cytometer was not used... this would have most likely had produced more accurate results as compared to microscope observations. Regarding the scope is not clear how the % of viable cells was calculated nor how representative your sample was. What was N? how many fields of view (FOVs) were counted per slide or well? In each of the independent cultures? If only one slide/independent culture counted, seems not enough to me. Not clear either which statistical analyses has been carried out for this? Epifluorescence (or any microscopically quantification) can lead to artefactual data unless N is large enough (not sure this is

the case, and clarification is needed), or other intercalibrating method is used to contrast with numbers, such as flow cytometry.

**See above reply to concern number 3**

6. Another question that is confusing to me is the lack of standardised methods for measuring cell performance. It is not clear enough why fluorescent probes were used with Tetraselmis, but not for Skeletonema nor for the natural community. Moreover, for Skeletonema fluorescence, authors do not specify which fluorescence was measured? Red 666nm? In which device was this measured? Green with probes? Again, in which device? Along the same line, the rationale for the natural community analysis it is not well understood, nor how the % of viable cells was also calculated. Table 3 and 4 are difficult to understand because it cannot be discriminated to which functional group each % belong, therefore, not sure I see the point for this. I'd like to point out that the methodology description does not suffice to understand how this experiment was performed. The ms. does not have an easy thread-line to be followed. Could it perhaps be better organised?

**The FDA/CMFDA was also used to count Skeletonema and natural phytoplankton. Fluorescence was measured at 645 nm.**

The following are inserted in 2.2 (wet test): " The cell density was determined by FDA and CMFDA double staining and fluorescence at 640 nM in SpectraMax iD3 microplates after approximately 24, 48 and 72 hours (±2h)."

And in 2.3. (Natural assemblage...): "The water quality and algal density was monitored daily in each beaker, using the same methods described in Chapter 2.1. Moreover, cell count and viability were quantified using the same protocol as in 2.1., with florescence measured at 645 nm."

**Ther were a few typos in table 3. They are now corrected. We hope that this makes the table easier to read. Moreover, table 4 and the paragraph discussing functional groups are omitted from the manuscript.**

7. The statistical approach used, does not seem appropriate. First, does data distribution meet the requirements for parametrical tests? Assuming so, T-tests do not capture the variability of the system you might have. Hence, most likely, significant differences are not well resolved, questioning the results.

I would suggest that a oneway ANOVA would be adequate since you have 5 levels of concentrations. Also, in those cases in which time is a continuous variable and not an end-point variable, the right approach could be a split-plot ANOVA in which the fixed factor would be the concentration and the repeated measures factor would be time.

**We thank the reviewer for pointing out the requirements for parametric tests. We have log transformed the values to obtain similar variation between the treatments at day 6. Our choice to use the T-Test instead of a repeated measures ANOVA, was based mainly on missing cells in the time series and the zero values just after exposure to  $CA(OH)_2$  creates problems related to normal distribution. However, we think that the T-test is appropriate for investigating differences between the treatments on day 6. Regarding the comment that this approach questions the results, we do not fully understand which part of the results this test questions? The dynamics of algae cultures exposed to the different**

parameters is depicted in Fig 2, and the T-Test clearly show that there are significant treatment effects after six days after exposure (p<0.001).

8. In my opinion the discussion is shallow. It does not get deep insight on explaining the data, nor debating them. Not enough quality for a discussion I'm afraid. The same applies to the conclusions, which are merely descriptive.

*# "We believe that the inclusion of Section 4.1, which discusses the model and experimental design, will enhance the clarity of our conclusions. Still, generally, we hold the view that the degree to which our findings should be contextualized and contrasted with other studies is appropriate. Furthermore, we would like to underscore that reviewer 2 has not presented any concerns pertaining to the discussion or conclusions.*